# HIDDEN MARKOV MIXTURE OF GAUSSIAN PROCESS FUNCTIONAL REGRESSION: UTILIZING MULTI-SCALE STRUCTURE FOR TIME-SERIES FORECASTING

## ABSTRACT

The mixture of Gaussian process functional regressions (GPFRs) assumes that there are a batch of time-series or sample curves which are generated by independent random processes with different temporal structures. However, in the real situations, these structures are actually transferred in a random manner from a long time scale. Therefore, the assumption of independent curves is not true in practice. In order to get rid of this limitation, we propose the hidden Markov based GPFR mixture model (HM-GPFR) by describing these curves with both fine and coarse level temporal structures. Specifically, the temporal structure is described by the Gaussian process model at the fine level and hidden Markov process at the coarse level. The whole model can be regarded as a random process with state switching dynamics. To further enhance the robustness of the model, we also give a priori to the model parameters and develop Bayesian hidden Markov based GPFR mixture model (BHM-GPFR). Experimental results demonstrate that the proposed methods have both high prediction accuracy and good interpretability.

## 1 INTRODUCTION

The time-series considered in this paper has the multi-scale structure: the *coarse level* and the *fine level*. We have observations $(\boldsymbol{y}_1, \ldots, \boldsymbol{y}_T)$ where each $\boldsymbol{y}_t = (y_{t,1}, \ldots, y_{t,L})$ itself is a time-series of length $L$. The whole time-series is arranged as

$$y_{1,1}, y_{1,2}, \ldots, y_{1,L}, \ y_{2,1}, y_{2,2}, \ldots, y_{2,L}, \ \ldots, y_{T,1}, y_{T,2}, \ldots, y_{T,L}. \qquad (1)$$

The subscripts of $\{\boldsymbol{y}_t\}_{t=1}^T$ are called coarse level indices, while the subscripts of $\{y_{t,i}\}_{i=1}^L$ are called fine level indices. Throughout this paper, we take the electricity load dataset as a concrete example. The electricity load dataset consists of $T = 365$ consecutive daily records, and in each day there are $L = 96$ samples recorded every quarter-hour. In this example, the coarse level indices denote "day", while the fine level indices correspond to the time resolution of 15 minutes. The aim is to forecast both short-term and long-term electricity loads based on historical records. There may be partial observations $y_{T+1,1}, \ldots, y_{T+1,M}$ with $M < L$, so the entire observed time-series has the form

$$y_{1,1}, y_{1,2}, \ldots, y_{1,L}, \ y_{2,1}, y_{2,2}, \ldots, y_{2,L}, \ \ldots, y_{T,1}, y_{T,2}, \cdots, \ y_{T,L}, y_{T+1,1}, \ldots, y_{T+1,M} \cdot \qquad (2)$$

The task is to predict future response $y_{t_*,i_*}$ where $t_* \geq T + 1, 1 \leq i_* \leq L$ are positive integers.

The coarse level and fine level provide different structural information about the data generation process. In the coarse level, each $\boldsymbol{y}_t$ can be regarded as a time-series, and there is certain cluster structure (Shi & Wang, 2008; Wu & Ma, 2018) underlying these time-series $\{\boldsymbol{y}_t\}_{t=1}^T$: we can divide $\{\boldsymbol{y}_t\}_{t=1}^T$ into groups such that time-series within each group share a similar evolving trend. Back to the electricity load dataset, such groups correspond to different electricity consumption patterns. We use $z_t$ to denote the cluster label of $\boldsymbol{y}_t$. In the fine level, observations $\{y_{t,i}\}_{i=1}^L$ can be regarded as a realization of a stochastic process, and the properties of the stochastic process are determined by the cluster label $z_t$.

The mixture of Gaussian processes functional regression (mix-GPFR) model (Shi & Wang, 2008; Shi & Choi, 2011) is powerful for analyzing functional data or batch data, and it is applicable to the multi-scale time-series forecasting task. Mix-GPFR assumes there are $K$ Gaussian processes

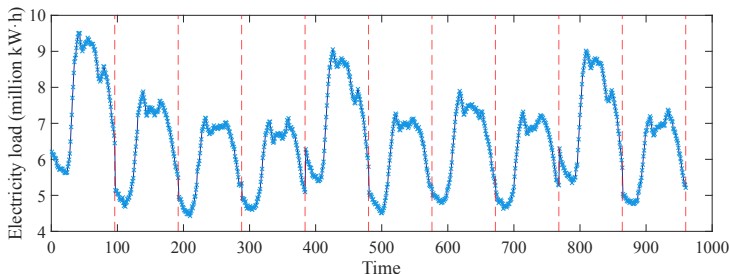

Figure 1: An illustration of multi-scale time-series.

functional regression (GPFR) (Shi et al., 2007) components, and associated with each $\boldsymbol{y}_t$ there is a latent variable $z_t$ indicating $\boldsymbol{y}_t$ is generated by which GPFR component. Since GPFR is good at capturing temporal dependency, this model successfully utilizes the structure information in the fine level. However, the temporal information in the coarse level is totally ignored since mix-GPFR assumes $\{z_t\}_{t=1}^T$ are i.i.d. .

In this work, we propose to model the temporal dependency in the coarse level by the hidden Markov model, which characterizes the switching dynamics of $z_1, \ldots, z_T$ by the transition probability matrix. We refer to the proposed model as HM-GPFR. Mix-GPFR is able to effectively predict $y_{T+1,M+1}, \ldots, y_{T+1,L}$ when $M > 0$. To predict the responses $y_{T+1,i_*}$, we must determine the cluster label $z_{T+1}$ based on observations $y_{T+1,1}, \ldots, y_{T+1,M}$, otherwise we do not know $\boldsymbol{y}_{T+1}$ is governed by which evolving pattern. If there is no observation at day $T + 1$ (i.e., $M = 0$), then mix-GPFR fails to identify the stochastic process that generates $\boldsymbol{y}_{T+1}$. For the same reason, mix-GPFR is not suitable for long-term forecasting ($t_* > T + 1$). On the other hand, HM-GPFR is able to infer $z_{t_*}$ for any $t_*$ based on the transition probabilities of the hidden Markov model even $M = 0$. Therefore, HM-GPFR makes use of coarse level temporal information and solves the cold start problem in mix-GPFR. Besides, when a new day's records $\boldsymbol{y}_{T+1}$ have been fully observed, one needs to re-train a mix-GPFR model to utilize $\boldsymbol{y}_{T+1}$, while HM-GPFR can adjust the parameters incrementally without retraining the model.

## 2 RELATED WORKS

Gaussian process (Rasmussen & Williams, 2006) is a powerful non-parametric Bayesian model. In (Girard et al., 2002; Brahim-Belhouari & Bermak, 2004; Girard & Murray-Smith, 2005), GP has been applied for time-series forecasting. Shi *et al.* proposed the GPFR model to process batch data (Shi et al., 2007). To effectively model multi-modal data, the mixture structure is further introduced to GPFR and the mix-GPFR model was proposed (Shi & Wang, 2008; Shi & Choi, 2011). In (Wu & Ma, 2018; Li et al., 2019; Cao et al., 2021), GP related methods for electricity load prediction have been evaluated thoroughly. However, in these works daily records are treated as i.i.d. samples, and the temporal information in the coarse level is ignored.

Multi-scale time-series was proposed in (Ferreira et al., 2006; Ferreira & Lee, 2007b;a), and further developments in this direction have been achieved in recent years. The time-series considered in this work is different from the multi-scale time-series since at the coarse level there is no aggregated observation from the samples at the fine level. In this paper, we mainly emphasize the multi-scale structure of the time-series.

## 3 PRELIMINARIES

### 3.1 HIDDEN MARKOV MODEL

For a sequence of observations $\{\boldsymbol{y}_t\}_{t=1}^T$, the hidden Markov model (HMM) (Rabiner & Juang, 1986; Elliott et al., 2008) assumes there is a hidden state variable $z_t$ associated with $\boldsymbol{y}_t$. The sequence of hidden states $\{z_t\}_{t=1}^T$ forms a homogeneous Markov process. Usually, $\{z_t\}_{t=1}^T$ are categorical variables taking values in $\{1, \ldots, K\}$, and the transition dynamics is governed by $\mathbb{P}(z_t = l | z_{t-1} =$

$k) = P_{kl}$. There are $K$ groups of parameters $\{\boldsymbol{\theta}_k\}_{k=1}^K$, and $z_t = k$ indicates that the observation $\boldsymbol{y}_t$ is generated by $\mathbb{P}(\boldsymbol{y}; \boldsymbol{\theta}_k)$. The goal of learning is to identify the parameters and infer the posterior distribution of hidden states $\{z_t\}_{t=1}^T$. Usually, the Baum-Welch algorithm (Baum & Petrie, 1966; Baum et al., 1970) is utilized to learn the HMM, which can be regarded as a specifically designed EM algorithm based on the forward-backward algorithm. Once the model has been trained, we are able to simulate future behavior of the system.

## 3.2 GAUSSIAN PROCESS FUNCTIONAL REGRESSIONS

Gaussian process is a stochastic process that any finite-dimensional distribution of samples is a multivariate Gaussian distribution. The property of a Gaussian process is determined by the mean function and the covariance function. We write the mean function as $\mu(\cdot)$ and the covariance function as $c(\cdot, \cdot)$. Suppose that we have a dataset $\mathcal{D} = \{(x_i, y_i)\}_{i=1}^L$. The relationship between input and output is connected by a function $\mathscr{Y}$, i.e., $\mathscr{Y}(x_i) = y_i$. Let $\boldsymbol{x} = [x_1, x_2, \ldots, x_L]^T$, $\boldsymbol{y} = [y_1, y_2, \ldots, y_L]^T$, then we assume $\boldsymbol{y}|\boldsymbol{x} \sim \mathcal{N}(\boldsymbol{\mu}, \mathbf{C})$ where $\boldsymbol{\mu} = [\mu(x_1), \mu(x_2), \ldots, \mu(x_L)]^T$ and $\mathbf{C}_{ij} = c(x_i, x_j)$. In machine learning, the mean function and the covariance function are usually parameterized. Here, we use the squared exponential covariance function (Rasmussen & Williams, 2006; Shi & Choi, 2011; Wu & Ma, 2018) $c(x_i, x_j; \boldsymbol{\theta}) = \theta_1^2 \exp\left(-\theta_2^2 \frac{(x_i - x_j)^2}{2}\right) + \theta_3^2 \delta_{ij}$, where $\delta_{ij}$ is the Kronecker delta function and $\boldsymbol{\theta} = [\theta_1, \theta_2, \theta_3]$. The mean function is modeled as a linear combination of B-spline basis functions (Shi et al., 2007; Shi & Choi, 2011). Suppose that we have $D$ B-spline basis functions $\{\phi_d(x)\}_{d=1}^D$. Let $\mu(x) = \sum_{d=1}^D b_d \phi_d(x)$ and $\boldsymbol{\Phi}$ be an $L \times D$ matrix with $\boldsymbol{\Phi}_{id} = \phi_d(x_i)$, $\boldsymbol{b} = [b_1, b_2, \ldots, b_D]^T$, then $\boldsymbol{y}|\boldsymbol{x} \sim \mathcal{N}(\boldsymbol{\Phi}\boldsymbol{b}, \mathbf{C})$. From the function perspective, this model can be denoted as $\mathscr{Y}(x) \sim \mathcal{GPFR}(x; \boldsymbol{b}, \boldsymbol{\theta})$.

We can use the Gaussian process to model the multi-scale time-series considered in this paper, and the key-point is transform the multi-scale time-series to a batch dataset. For each coarse level index $t$, we can construct a dataset $\mathcal{D}_t = \{(x_{t,i}, y_{t,i})\}_{i=1}^L$, where $x_{t,i}$ is the sampling time of $i$-th sample in time-series $\boldsymbol{y}_t$. Let $\mathscr{Y}_t$ be the function underlying dataset $\mathcal{D}$, i.e., $\mathscr{Y}_t(x_{t,i}) = y_{t,i}$, then these $\{\mathcal{D}_t\}_{t=1}^T$ can be regarded as independent realizations of a GPFR, which assumes $\mathscr{Y}_t(x) \overset{\text{i.i.d.}}{\sim} \mathcal{GPFR}(x; \boldsymbol{b}, \boldsymbol{\theta})$. Without loss of generality, we may assume $x_{t,i} = i$, and thus $\boldsymbol{\Phi}_{id} = \phi_d(i)$, $\mathbf{C}_{ij} = c(i, j; \boldsymbol{\theta})$ do not depend on the coarse level index $t$. Therefore, it is equivalent to assume $\{\boldsymbol{y}_t\}_{t=1}^T$ are independently and identically distributed as $\mathcal{N}(\boldsymbol{\Phi}\boldsymbol{b}, \mathbf{C})$. To learn the parameters $\boldsymbol{b}$ and $\boldsymbol{\theta}$, we apply the Type-II maximum likelihood estimation technique (Rasmussen & Williams, 2006; Shi & Choi, 2011).

As for prediction, given a new record $\{(x_{t_*,i}, y_{t_*,i})\}_{i=1}^M$ and we want to predict the corresponding output $y_{t_*,i_*}$ at $x_{t_*,i_*}$ where $M < i_* \leq L$, from the definition of Gaussian process we immediately know that $y_{t_*,i_*}$ also obeys a Gaussian distribution (Rasmussen & Williams, 2006). Let

$$\boldsymbol{x}_* = [x_{t_*,1}, \ldots, x_{t_*,M}]^T, \boldsymbol{y}_* = [y_{t_*,1}, \ldots \quad , \quad y_{t_*,M}]^T, \tag{3}$$

$$\boldsymbol{\mu}_* = [\mu(x_{t_*,1}) \quad , \quad \ldots, \mu(x_{t_*,M})]^T, [\mathbf{C}_*]_{ij} = c(x_{t_*,i}, x_{t_*,j}), \tag{4}$$

then the mean of $y_{t_*,i_*}$ is $\mu(x_{t_*,i_*}) + \mathbf{c}(x_{t_*,i_*}, \boldsymbol{x}_*)\mathbf{C}_*^{-1}(\boldsymbol{y}_* - \boldsymbol{\mu}_*)$, and the variance of $y_{t_*,i_*}$ is $c(x_{t_*,i_*}, x_{t_*,i_*}) - \mathbf{c}(x_{t_*,i_*}, \boldsymbol{x}_*)\mathbf{C}_*^{-1}\mathbf{c}(\boldsymbol{x}_*, x_{t_*,i_*})$. Note that if $M = 0$, the prediction is simply given by $\mathcal{N}(\mu(x_{t_*,i_*}), c(x_{t_*,i_*}, x_{t_*,i_*}))$, which equals to the *prior* distribution of $y_{t_*,i_*}$ and fails to utilize the temporal dependency with recent observations. In the electricity load prediction example, this means we can only effectively predict a new day's electricity loads when we already have the first few observations of this day. In practice, however, it is very common to predict a new day's electricity loads from scratch.

## 3.3 THE MIXTURE OF GAUSSIAN PROCESS FUNCTIONAL REGRESSIONS

GPFR implicitly assumes that all $\{\boldsymbol{y}_t\}_{t=1}^T$ are generated by the same stochastic process, which is not the case in practice. In real applications, they may be generated from different signal sources, thus a single GPFR is not flexible enough to model all the time series, especially when there are a variety of evolving trends. Take the electricity load dataset for example, the records corresponding to winter and summer are very likely to have significantly different trends and shapes. To solve this problem, Shi *et al.*() suggested to introduce the mixture structure to GPFR, and proposed the mixture of Gaussian process functional regressions (mix-GPFR). In mix-GPFR, there are $K$ GPFR

components with different parameters $\{\boldsymbol{b}_k, \boldsymbol{\theta}_k\}_{k=1}^{K}$, and the mixing proportion of the $k$-th GPFR component is $\pi_k$. Intuitively, there are $K$ different signal sources or evolving patterns in mix-GPFR to describe temporal data with different temporal properties. For each $\boldsymbol{y}_t$, there is an associated latent indicator variable $z_t \in \{1, 2, \ldots, K\}$, and $z_t = k$ indicates $\boldsymbol{y}_t$ is generated by the $k$-th GPFR component. The generation process of mix-GPFR is

$$
\begin{aligned}
z_t &\overset{\text{i.i.d.}}{\sim} \text{Categorical}(\pi_1, \pi_2, \ldots, \pi_K), \\
\mathscr{Y}_t(x)|z_t = k &\sim \mathcal{GPFR}(x; \boldsymbol{b}_k, \boldsymbol{\theta}_k).
\end{aligned}
\tag{5}
$$

Let $\mathbf{C}_k \in L \times L$ be the covariance matrix calculated by $\boldsymbol{\theta}_k$, i.e., $[\mathbf{C}_k]_{ij} = c(i, j; \boldsymbol{\theta}_k)$, then the above equation is equivalent to $\boldsymbol{y}_t \sim \mathcal{N}(\boldsymbol{\Phi}\boldsymbol{b}_k, \mathbf{C}_k)$.

Due to the existence of latent variables, the parameter learning of mix-GPFR involves the EM algorithm (Dempster et al., 1977; Shi & Wang, 2008). As for prediction, $K$ GPFR components of mix-GPFR first make predictions individually, then we weight these predictions based on the posterior probability $\mathbb{P}(z_{t_*} = k|\boldsymbol{y}_{t_*}; \boldsymbol{b}_k, \boldsymbol{\theta}_k)$. Note that if $M = 0$, then $\mathbb{P}(z_{t_*} = k|\boldsymbol{y}_{t_*}; \boldsymbol{b}_k, \boldsymbol{\theta}_k) = \pi_k$, which equals to the mixing proportions and also fails to utilize recent observations. Therefore, mix-GPFR also suffers from the cold start problem.

## 4    PROPOSED METHODS

### 4.1    HIDDEN MARKOV BASED GAUSSIAN PROCESS FUNCTIONAL REGRESSION MIXTURE MODEL

Similar to mix-GPFR, the hidden Markov based Gaussian process functional regression mixture model also assumes the time-series is generated by $K$ signal sources. The key difference is that the signal source may switch between consecutive observations in the time resolution of the coarse level. The temporal structure in the coarse level is characterized by the transition dynamics of $\{z_t\}_{t=1}^{T}$, and the temporal dependency in the fine level is captured by Gaussian processes. Precisely,

$$
\begin{aligned}
z_1 &\sim \text{Categorical}(\pi_1, \pi_2, \ldots, \pi_K), \\
\mathbb{P}(z_t = l|z_{t-1} = k) &= P_{kl}, \, t = 2, 3, \ldots, T \\
\mathscr{Y}_t(x)|z_t = k &\sim \mathcal{GPFR}(x; \boldsymbol{b}_k, \boldsymbol{\theta}_k), \, t = 1, 2, \ldots, T.
\end{aligned}
\tag{6}
$$

Here, $\boldsymbol{\pi} = [\pi_1, \pi_2, \ldots, \pi_K]$ is the initial state distribution, and $\mathbf{P} = [P_{kl}]_{K \times K}$ is the transition probability matrix. We refer to this model as HM-GPFR. In GPFR and mix-GPFR, the observations $\{\boldsymbol{y}_t\}_{t=1}^{T}$ are modeled as independent and exchangeable realizations of stochastic processes, thus the temporal structure in the coarse level is destroyed. However, in HM-GPFR, consecutive $\boldsymbol{y}_{t-1}, \boldsymbol{y}_t$ are connected by the transition dynamics of their corresponding latent variables $z_{t-1}, z_t$, which is more suitable for time-series data. For example, if today's electricity loads are very high, then it is unlikely that tomorrow's electricity loads are extremely low.

The learning algorithm for HM-GPFR is based on the EM algorithm, and we derive the algorithm in the appendix. After the parameters have been learned, we assign the latent variable $\hat{z}_t = \arg\max_{k=1,\ldots,K} \gamma_t(k)$ and regard $\{\hat{z}_t\}_{t=1}^{T}$ as deterministic. For prediction, we consider two cases: $t_* = T + 1$ and $t_* > T + 1$. When $t_* = T + 1$, the latent variable $z_{T+1}$ is determined by both the conditional transition probability $z_{T+1}|\hat{z}_T$ and partial observations $\boldsymbol{y}_{T+1}$. More precisely, suppose $\hat{z}_T = l$, then

$$
\omega_k = \mathbb{P}(z_{T+1} = k|\mathcal{T}, \boldsymbol{y}_{T+1}, \hat{z}_T = l; \hat{\boldsymbol{\Theta}}) \propto \hat{P}_{lk} \mathcal{N}(\boldsymbol{y}_{T+1}; \boldsymbol{\Phi}[1:M, :]\hat{\boldsymbol{b}}_k, \mathbf{C}[1:M, 1:M]),
\tag{7}
$$

where the square brackets denote slicing operation. If $M = 0$, then $\omega_k = \hat{P}_{lk}$ is determined by the last hidden state and transition dynamics, which is more accurate than mix-GPFR. Suppose the prediction of the $k$-th component is $y_*^{(k)}$, then the final prediction is given by $\sum_{k=1}^{K} \omega_k y_*^{(k)}$.

We next consider the case $t_* > T + 1$, the main difference is the posterior distribution of $z_{t_*}$. In this case, we need to use the transition probability matrix recursively. First, we calculate the distribution of $z_{T+1}$ according to Equation (7). Then by the Markov property, we know

$$
\omega_k = \mathbb{P}(z_{t_*} = k|\mathcal{T}, \boldsymbol{y}_{T+1}, z_T = l; \hat{\boldsymbol{\Theta}}) \propto \sum_{m=1}^{K} \mathbb{P}(z_{T+1} = m|\mathcal{T}, \boldsymbol{y}_{T+1}, \hat{z}_T = l; \boldsymbol{\Theta})[\hat{\mathbf{P}}^{t_* - T - 1}]_{mk}.
\tag{8}
$$

The final prediction is also given by $\sum_{k=1}^{K} \omega_k y_*^{(k)} = \sum_{k=1}^{K} \omega_k \boldsymbol{\Phi}[i_*, :]\boldsymbol{b}_k$.

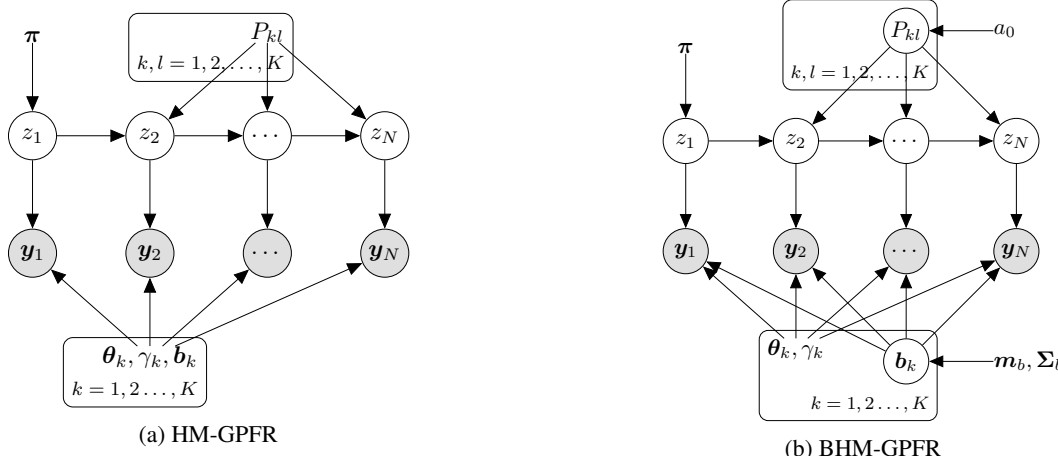

Figure 2: Probabilistic graphical models of HM-GPFR and BHM-GPFR.

## 4.2 BAYESIAN HIDDEN MARKOV BASED GAUSSIAN PROCESS FUNCTIONAL REGRESSION MIXTURE MODEL

One drawback of HM-GPFR is that there are too many parameters and thus has the risk of overfitting. In this section, we further develop a fully Bayesian treatment of HM-GPFR. We place a Gaussian prior $\mathcal{N}(\mathbf{m}_b, \boldsymbol{\Sigma}_b)$ on the coefficients of B-spline functions $\{\boldsymbol{b}_k\}_{k=1}^{K}$. For the transition probabilities, let $\mathbf{p}_k = [P_{k1}, P_{k2}, \ldots, P_{kK}]^{\mathrm{T}}$ be the probabilities from state $k$ to other states, then we assume $\mathbf{p}_k$ obeys a Dirichlet prior $\mathrm{Dir}(a_0, \ldots, a_0)$. The generation process of Bayesian HM-GPFR is

$$
\begin{aligned}
\boldsymbol{b}_k &\sim \mathcal{N}(\mathbf{m}_b, \boldsymbol{\Sigma}_b), \; k = 1, 2, \ldots, K \\
\boldsymbol{p}_k &\sim \mathrm{Dir}(a_0, \ldots, a_0), \; k = 1, 2, \ldots, K \\
z_1 &\sim \mathrm{Categorical}(\pi_1, \pi_2, \ldots, \pi_K), \\
\mathbb{P}(z_t = l | z_{t-1} = k) &= P_{kl}, \; t = 2, 3, \ldots, T \\
\mathscr{Y}_t(x) | z_t = k &\sim \mathcal{GPFR}(x; \boldsymbol{b}_k, \boldsymbol{\theta}_k), \; t = 1, 2, \ldots, T.
\end{aligned}
\tag{9}
$$

The detailed learning algorithm is presented in the appendix. After learning, we set the latent variables to their maximum a posteriori (MAP) estimates $\hat{\boldsymbol{\Omega}}$. Specifically, $\hat{\boldsymbol{b}}_k = \mathbf{m}_k$, $\hat{P}_{kl} = \frac{a_{kl}}{\sum_{m=1}^{K} a_{km}}$, $\hat{z}_t = \arg\max_{k=1,2,\ldots,K} \gamma_t(k)$. The rest of prediction is the same as HM-GPFR.

## 5 EXPERIMENTAL RESULTS

### 5.1 EXPERIMENT SETTINGS

In this section, we use the electricity load dataset issued by the State Grid of China for a city in northwest China. The dataset records electricity loads every 15 minutes, thus there are 96 records per day. Using the electricity load records of 2010 for training, we predict the subsequent $S$-step electricity loads in a time-series prediction fashion, where $S = 1, 2, 3, 4, 5, 10, 20, 30, 50, 80, 100, 200, 500, 1000$. This setting allows both short-term and long-term predictions to be evaluated. For a more comprehensive and accurate assessment of the performance, we roll the time series by 100 rounds. Based on the electricity loads of 2010, the $r$-th round also puts the first $(r-1)$ records of 2011 into the training set. In each round, we predict the subsequent $S$-step electricity loads. In $r$-th round, suppose the ground-truths are $y_1, y_2, \ldots, y_S$ and the predictions are $\hat{y}_1, \hat{y}_2, \ldots, \hat{y}_S$, we use the Mean Absolute Percentage Errors (MAPEs) to evaluate the prediction results. Specifically, $\mathrm{MAPE}_r = \frac{1}{S} \sum_{s=1}^{S} \frac{|y_s - \hat{y}_s|}{|y_s|}$. For overall evaluation, we report the average of 100 MAPEs to obtain $\mathrm{MAPE} = \frac{1}{100} \sum_{r=1}^{100} \mathrm{MAPE}_r$. Since the algorithms are influenced by randomness, we repeat the algorithms for 10 runs and report the average results.

We compare HM-GPFR and BHM-GPFR with other time-series forecasting methods. Specifically,

Table 1: MAPE of various methods on the electricity loads dataset under different step lengths and parameter settings.

| Method | Parameter | Step length $S$ | | | | | | | | | | | | | | |
|---|---|---|---|---|---|---|---|---|---|---|---|---|---|---|---|---|
| | | 1 | 2 | 3 | 4 | 5 | 10 | 20 | 30 | 50 | 80 | 100 | 200 | 300 | 500 | 1000 |
| AR | $L=4$ | 1.02% | 1.36% | 1.75% | 2.13% | 2.53% | 4.37% | 6.95% | 8.88% | 11.79% | 13.92% | 15.0% | 18.06% | 17.82% | 16.94% | 16.46% |
| | $L=8$ | 1.01% | 1.36% | 1.75% | 2.14% | 2.55% | 4.47% | 7.0% | 8.7% | 11.28% | 13.36% | 14.5% | 17.8% | 17.64% | 16.83% | 16.4% |
| | $L=12$ | 1.01% | 1.35% | 1.74% | 2.13% | 2.54% | 4.46% | 6.96% | 8.63% | 11.17% | 13.23% | 14.38% | 17.74% | 17.6% | 16.81% | 16.39% |
| MA | $L=4$ | 3.39% | 5.74% | 8.03% | 9.94% | 11.35% | 14.08% | 15.13% | 15.2% | 15.66% | 16.39% | 16.97% | 19.05% | 18.48% | 17.33% | 16.65% |
| | $L=8$ | 2.23% | 3.5% | 4.74% | 5.81% | 6.83% | 11.11% | 13.65% | 14.21% | 15.07% | 16.01% | 16.67% | 18.9% | 18.38% | 17.27% | 16.62% |
| | $L=12$ | 1.83% | 2.76% | 3.66% | 4.46% | 5.21% | 8.61% | 12.32% | 13.33% | 14.54% | 15.68% | 16.41% | 18.77% | 18.29% | 17.22% | 16.59% |
| ARMA | $L=4$ | 1.01% | 1.34% | 1.73% | 2.12% | 2.52% | 4.42% | 6.93% | 8.6% | 11.22% | 13.18% | 14.31% | 17.64% | 17.54% | 16.77% | 16.38% |
| | $L=8$ | 1.01% | 1.34% | 1.72% | 2.09% | 2.48% | 4.34% | 6.87% | 8.52% | 11.12% | 13.05% | 14.13% | 17.5% | 17.44% | 16.71% | 16.34% |
| | $L=12$ | 1.02% | 1.36% | 1.76% | 2.14% | 2.55% | 4.39% | 6.8% | 8.4% | 10.99% | 12.89% | 13.93% | 17.31% | 17.3% | 16.62% | 16.3% |
| ARIMA | $L=4$ | 0.98% | 1.34% | 1.74% | 2.14% | 2.57% | 4.58% | 7.27% | 9.08% | 11.99% | 14.43% | 15.34% | 18.65% | 18.67% | 17.95% | 17.65% |
| | $L=8$ | 1.01% | 1.36% | 1.75% | 2.14% | 2.57% | 4.56% | 7.24% | 9.2% | 12.48% | 14.53% | 15.09% | 18.67% | 18.79% | 18.2% | 18.37% |
| | $L=12$ | 1.01% | 1.4% | 1.82% | 2.24% | 2.68% | 4.93% | 8.64% | 11.92% | 18.41% | 22.65% | 21.83% | 24.05% | 24.24% | 24.2% | 29.52% |
| SARMA | $L=4$ | 0.83% | 1.08% | 1.33% | 1.55% | 1.76% | 2.66% | 4.06% | 5.15% | 6.38% | 7.57% | 8.67% | 10.69% | 9.96% | 7.62% | 7.62% |
| | $L=8$ | 0.83% | 1.08% | 1.32% | 1.55% | 1.76% | 2.67% | 4.04% | 5.12% | 6.35% | 7.54% | 8.64% | 10.67% | 9.93% | 7.58% | 7.58% |
| | $L=12$ | 0.82% | 1.07% | 1.3% | 1.52% | 1.72% | 2.62% | 4.06% | 5.16% | 6.31% | 7.17% | 8.11% | 10.55% | 10.09% | 7.86% | 7.83% |
| LSTM | $L=4$ | 12.89% | 12.9% | 12.91% | 12.97% | 13.04% | 13.55% | 14.56% | 15.16% | 16.24% | 16.99% | 17.25% | 19.48% | 19.01% | 17.88% | 17.28% |
| | $L=12$ | 12.39% | 12.32% | 12.32% | 12.35% | 12.39% | 12.78% | 13.9% | 14.83% | 16.38% | 17.38% | 17.42% | 19.77% | 19.39% | 18.27% | 17.73% |
| | $L=24$ | 11.48% | 11.43% | 11.43% | 11.46% | 11.5% | 11.81% | 12.69% | 13.49% | 14.73% | 15.72% | 16.28% | 18.97% | 18.8% | 17.83% | 17.44% |
| | $L=48$ | 10.1% | 10.11% | 10.11% | 10.15% | 10.2% | 10.49% | 11.22% | 11.96% | 12.94% | 13.09% | 13.56% | 16.53% | 17.52% | 17.98% | 18.57% |
| FNN | $L=4$ | 0.96% | 1.29% | 1.64% | 1.94% | 2.27% | 3.99% | 6.21% | 8.13% | 11.56% | 14.4% | 15.49% | 18.71% | 18.71% | 17.87% | 17.61% |
| | $L=12$ | 0.85% | 1.1% | 1.37% | 1.62% | 1.88% | 3.13% | 5.38% | 7.25% | 9.94% | 13.24% | 14.87% | 20.44% | 20.72% | 19.91% | 19.81% |
| | $L=24$ | 0.85% | 1.07% | 1.27% | 1.43% | 1.6% | 2.39% | 3.94% | 5.38% | 7.43% | 10.1% | 11.54% | 14.57% | 15.27% | 15.62% | 17.87% |
| | $L=48$ | 0.85% | 1.0% | 1.15% | 1.28% | 1.39% | 1.99% | 3.21% | 4.12% | 5.49% | 7.62% | 8.93% | 10.26% | 9.42% | 7.72% | 8.34% |
| SVR | $L=4$ | 0.98% | 1.33% | 1.71% | 2.05% | 2.43% | 4.16% | 5.85% | 7.81% | 10.82% | 14.1% | 15.07% | 18.94% | 19.84% | 19.41% | 19.5% |
| | $L=12$ | 1.05% | 1.33% | 1.62% | 1.91% | 2.17% | 3.6% | 6.59% | 9.09% | 13.5% | 17.89% | 19.2% | 24.47% | 27.77% | 28.52% | 29.88% |
| | $L=24$ | 1.06% | 1.29% | 1.5% | 1.68% | 1.85% | 2.73% | 4.82% | 6.85% | 9.56% | 12.47% | 13.85% | 17.34% | 17.83% | 17.56% | 18.33% |
| | $L=48$ | 1.25% | 1.46% | 1.64% | 1.8% | 1.95% | 2.66% | 4.1% | 5.27% | 7.9% | 11.33% | 13.05% | 12.39% | 9.87% | 8.45% | 8.07% |
| EGPM | $L=4, K=3$ | 0.97% | 1.29% | 1.65% | 1.98% | 2.33% | 4.05% | 6.42% | 7.54% | 10.22% | 13.49% | 15.06% | 18.17% | 17.97% | 17.15% | 16.81% |
| | $L=4, K=5$ | 0.97% | 1.28% | 1.64% | 1.97% | 2.32% | 4.03% | 6.38% | 7.53% | 10.18% | 13.45% | 15.04% | 18.18% | 17.97% | 17.16% | 16.83% |
| | $L=4, K=10$ | 0.97% | 1.29% | 1.65% | 1.98% | 2.33% | 4.04% | 6.42% | 7.57% | 10.23% | 13.5% | 15.07% | 18.19% | 17.98% | 17.16% | 16.82% |
| | $L=12, K=3$ | 0.93% | 1.19% | 1.49% | 1.77% | 2.08% | 3.65% | 5.92% | 8.16% | 11.44% | 14.12% | 15.35% | 18.99% | 19.32% | 18.78% | 18.44% |
| | $L=12, K=5$ | 0.92% | 1.18% | 1.47% | 1.76% | 2.06% | 3.63% | 5.89% | 8.17% | 11.45% | 14.13% | 15.35% | 19.02% | 19.32% | 18.79% | 18.47% |
| | $L=12, K=10$ | 0.95% | 1.21% | 1.51% | 1.79% | 2.1% | 3.67% | 5.92% | 8.13% | 11.4% | 14.15% | 15.39% | 19.05% | 19.32% | 18.77% | 18.42% |
| | $L=24, K=3$ | 0.94% | 1.19% | 1.41% | 1.6% | 1.81% | 2.95% | 5.09% | 7.09% | 9.79% | 13.34% | 15.26% | 19.03% | 19.66% | 20.47% | 21.91% |
| | $L=24, K=5$ | 0.97% | 1.22% | 1.43% | 1.62% | 1.83% | 2.97% | 5.01% | 6.96% | 9.51% | 12.62% | 14.39% | 17.57% | 17.85% | 18.66% | 20.4% |
| | $L=24, K=10$ | 0.95% | 1.2% | 1.42% | 1.62% | 1.82% | 2.91% | 4.89% | 6.84% | 9.19% | 12.32% | 14.18% | 17.67% | 18.28% | 19.14% | 20.78% |
| | $L=48, K=3$ | 1.02% | 1.29% | 1.52% | 1.74% | 1.93% | 2.83% | 5.42% | 7.03% | 9.0% | 11.88% | 13.93% | 23.48% | 32.01% | 38.54% | 45.35% |
| | $L=48, K=5$ | 1.02% | 1.28% | 1.52% | 1.74% | 1.93% | 2.92% | 5.68% | 7.44% | 9.46% | 12.35% | 14.42% | 24.12% | 32.28% | 38.92% | 45.36% |
| | $L=48, K=10$ | 1.02% | 1.29% | 1.53% | 1.76% | 1.95% | 2.94% | 5.7% | 7.43% | 9.38% | 12.29% | 14.38% | 23.61% | 32.3% | 38.86% | 45.69% |
| mix-GPFR | $P=30, K=5$ | 0.82% | 0.97% | 1.12% | 1.25% | 1.39% | 2.09% | 3.37% | 4.24% | 5.75% | 7.94% | 9.19% | 10.67% | 9.65% | 7.19% | 7.24% |
| mixGPNM | $K=5$ | 0.78% | 0.94% | 1.11% | 1.26% | 1.4% | 2.16% | 3.47% | 4.34% | 5.85% | 8.02% | 9.27% | 10.71% | 9.67% | 7.2% | 7.25% |
| DPM-GPFR | $P=30$ | 0.83% | 0.91% | 0.97% | 1.03% | 1.09% | 1.4% | 2.09% | 2.61% | 3.38% | 4.14% | 4.8% | 10.15% | 12.35% | 12.26% | 12.81% |
| HM-GPFR | $P=30, K=5$ | 0.93% | 1.12% | 1.3% | 1.48% | 1.66% | 2.51% | 4.07% | 5.18% | 6.79% | 8.8% | 9.83% | 10.76% | 9.49% | 6.82% | 6.77% |
| BHM-GPFR | $P=30, K=5$ | 0.77% | 0.92% | 1.07% | 1.18% | 1.3% | 1.89% | 2.88% | 3.59% | 4.89% | 6.88% | 8.04% | 9.85% | 9.21% | 6.94% | 7.15% |

- Classical times-series forecasting methods: auto-regressive (AR), moving average (MA), auto-regressive moving average (ARMA), auto-regressive integrated moving average (ARIMA), seasonal auto-regressive moving average (SARMA).

- Machine learning methods: long short-term memory (LSTM), feedforward neural network (FNN), support vector regression (SVR), enhanced Gaussian process mixture model (EGPM).

- GPFR related methods: the mixture of Gaussian process functional regressions (mix-GPFR), the mixture of Gaussian processes with nonparametric mean functions (mix-GPNM), Dirichlet process based mixture of Gaussian process functional regressions (DPM-GPFR).

Detailed parameter settings of comparison methods are shown in the appendix. The main parameters in HM-GPFR and BHM-GPFR are the number of components $K$ and the number of B-spline basis functions $D$, and we set $K = 5, D = 30$.

## 5.2 Performance Evaluation and Model Explanation

The prediction results of various methods on the electricity load dataset are shown in table 1. From the table, we can see that the prediction accuracy of classical time-series forecast methods decreases significantly as we increase the prediction step. Among them, SARMA outperforms AR, MA, ARMA, and ARIMA, because SARMA takes the periodicity of data into consideration and can fit data more effectively. The results of machine learning methods LSTM, NN, SVR, and EGPM also have similar phenomena, that is, when $S$ is small, the prediction accuracy is high, and when $S$ is large, the prediction accuracy is low. This observation indicates that these methods are not suitable for long-term prediction. In addition, machine learning methods are also sensitive to the settings of parameters. For example, the results of FNN and SVR are better when $L = 4$, which is close to SARMA, while the long-term prediction accuracy of EGPM decreases significantly when $L$ is relatively large. It is challenging to appropriately set hyper-parameters in practice. When making

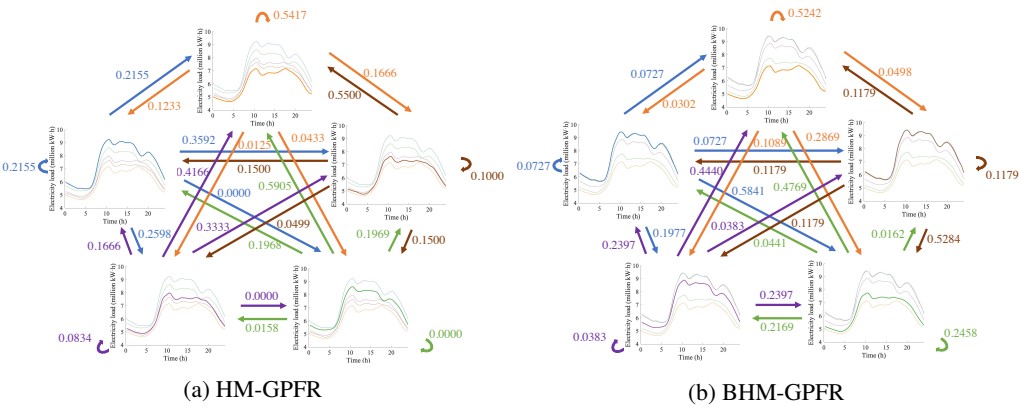

(a) HM-GPFR

(b) BHM-GPFR

Figure 3: Evolving law of electricity loads and transition dynamics learned by HM-GPFR and BHM-GPFR.

a long-term prediction, classical time-series prediction methods and machine learning methods need to recursively predict the subsequent values based on estimated values, which will cause the accumulation and amplification of errors. On the other hand, GPFR-related methods first make predictions according to the mean function, then finely correct these predictions based on observed data. The mean function part can better describe the evolution law of data, which enables us to historical information and structural information in data more effectively. Mix-GPFR, mix-GPNM, and DPM-GPFR obtain similar results in long-term prediction compared with SARMA, and can even achieve the best results in short-term prediction. This observation demonstrates the effectiveness of GPFR-related methods. However, these methods cannot deal with long-term prediction tasks well due to the "cold start" problem. Overall, the performances of the proposed HM-GPFR and BHM-GPFR are more comprehensive. For medium-term and short-term prediction, the results of HM-GPFR and BHM-GPFR are slightly worse than those of mix-GPFR, mix-GPNM, and DPM-GPFR, but they still enjoy significant advantages compared with other comparison methods. In terms of long-term forecasting, HM-GPFR and BHM-GPFR outperform mix-GPFR, mix-GPNM, and DPM-GPFR, which shows that considering the multi-scale temporal structure between daily electricity load time-series can effectively improve the accuracy of long-term forecasting. In addition, BHM-GPFR is generally better than HM-GPFR, which shows that giving prior distributions to the parameters and learning in a fully Bayesian way can further increase the robustness of the model and improve the prediction accuracy.

HM-GPFR and BHM-GPFR have strong interpretability. Specifically, the estimated values of hidden variables obtained after training $\{\hat{z}_i\}_{i=1}^n$ divide the daily electricity load records into $K$ categories according to the evolution law. Each evolution pattern can be represented by the mean function of GPFR component, and these evolution patterns transfer to each other with certain probabilities. The transfer law is characterized by the transfer probability matrix in the model. In fig. 3, we visualize the evolution patterns and transfer laws learned by HM-GPFR and BHM-GPFR. We call the evolution law corresponding to the mean function represented by the orange curve (at the top of the figure) mode 1, and call the five evolution modes as mode 1 to mode 5 respectively in clockwise order. Combined with the practical application background, some meaningful laws can be found according to the results of learned models. Examples are as follows:

- The electricity load of mode 1 is the lowest. Besides, mode 1 is relatively stable: when the system is in this evolution pattern, then it will stay in this state in the next step with a probability of about $0.5$. In case of state transition, the probability of transferring to the mode with second lowest load (mode 2 in Figure 3a and mode 3 in Figure 3b) is high, while the probability of transferring to the mode with highest load (mode 5 in Figure 3a and mode 2, mode 5 in Figure 3b) is relatively low;

- The evolution laws of mode 2 and mode 5 in fig. 3b are very similar, but the probabilities of transferring to other modes are different. From the perspective of electricity load alone, both of them can be regarded as the mode with the highest load. When the system is in the mode with the highest load (mode 5 in Figure 3a and mode 2, mode 5 in Figure 3b), the

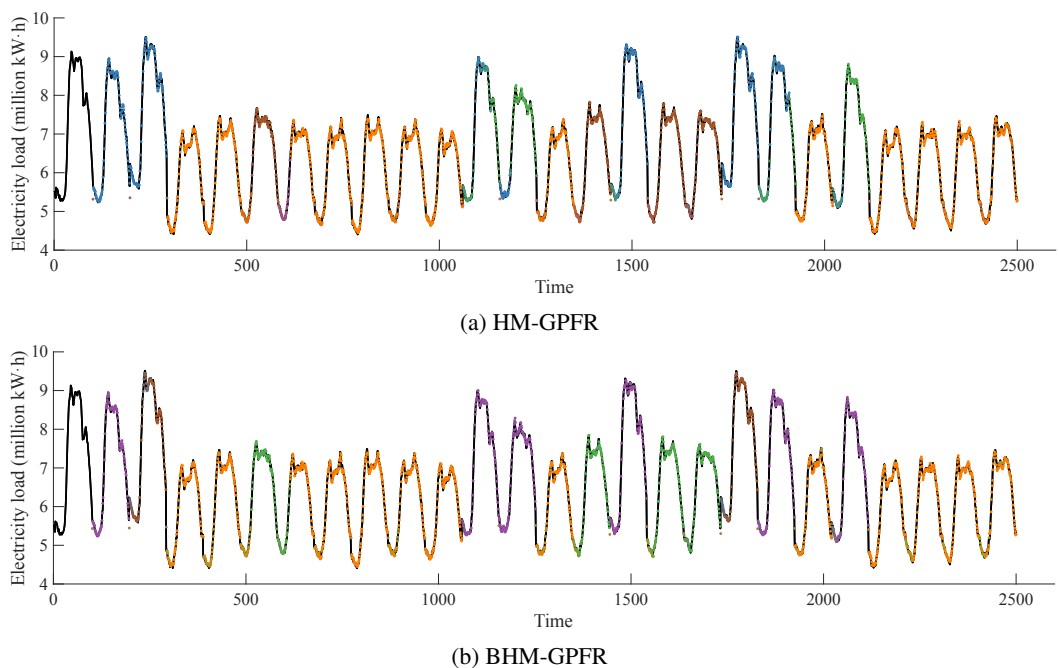

(a) HM-GPFR

(b) BHM-GPFR

Figure 4: One-step-ahead rolling prediction results of HM-GPFR and BHM-GPFR.

probability of remaining in this state in the next step is the same as that of transferring to the mode with the lowest (mode 1);

- When the system is in the mode with the second-highest load (mode 3 in fig. 3a and mode 4 in fig. 3b), the probability of remaining in this state in the next step is low, while the probabilities of transferring to the modes with the lowest load and the highest load are high.

These laws are helpful for us to understand the algorithm, have a certain guiding significance for production practice, and can also be further analyzed in combination with expert knowledge.

The case of $S = 1$ in table 1 is the most common in practical application, that is, one-step-ahead rolling forecast. As discussed in section 4.1, when making a rolling prediction, HM-GPFR and BHM-GPFR can dynamically adjust the model incrementally after collecting new data without retraining the model. The results of the one-step-ahead rolling prediction of HM-GPFR and BHM-GPFR on the electricity load dataset are shown in fig. 4. It can be seen that the predicted values of HM-GPFR and BHM-GPFR are very close to the ground-truths, indicating that they are effective for rolling prediction. In the figure, the color of each point is the weighted average of the colors corresponding to each mode in fig. 3 according to the weight $\omega_K$. Note that there are color changes in some electricity load curves in fig. 4a and fig. 4b. Taking the time-series in fig. 4a in the range of about 1100-1200 as an example, when there are few observation data on that day, HM-GPFR believes that the electricity load evolution pattern of that day is more likely to belong to mode 3. With the gradual increase of observation data, the model tends to think that the electricity load evolution pattern of that day belongs to mode 5, and then tends to mode 3 again. This shows that HM-GPFR and BHM-GPFR can timely adjust the value of $z_{i_*}$ according to the latest information during rolling prediction.

## 5.3 ABLATION STUDY

In this section, we mainly compare HM-GPFR, BHM-GPFR with mix-GPFR, mix-GPNM, and DPM-GPFR to explore the impact of introducing coarse-grained temporal structure on the prediction performance. The MAPEs reported in Table 2 are averaged with respect to $r = 1, \ldots, 100$, while in this section we pay special attention to the case of $r = 1$. In this case, the observed data is the electricity load records in 2010, and there is no partial observations on January 1, 2011 (*i.e.*, $M = 0$ in eq. (2)). Therefore, mix-GPFR, mix-GPNM, and DPM-GPFR will encounter the cold-start problem. Table 2 reports the MAPE of these methods at different prediction steps when $r = 1$. It can

Table 2: MAPE of GP related methods under the cold-start setting ($r = 1$).

| Method | Step length $S$ | | | | | | | | | | | | | | |
|---|---|---|---|---|---|---|---|---|---|---|---|---|---|---|---|
| | 1 | 2 | 3 | 4 | 5 | 10 | 20 | 30 | 50 | 80 | 100 | 200 | 300 | 500 | 1000 |
| mix-GPFR | 6.72% | 6.76% | 6.98% | 7.02% | 7.21% | 7.12% | 7.18% | 7.07% | 8.98% | 8.69% | 8.49% | 11.66% | 10.85% | 7.93% | 7.4% |
| mix-GPNM | 6.72% | 6.76% | 6.98% | 7.03% | 7.2% | 7.11% | 7.18% | 7.07% | 8.98% | 8.69% | 8.48% | 11.66% | 10.85% | 7.93% | 7.4% |
| DPM-GPFR | 11.79% | 11.81% | 12.05% | 12.14% | 12.35% | 12.41% | 12.66% | 12.53% | 13.75% | 12.97% | 12.68% | 15.35% | 11.93% | 8.4% | 6.41% |
| HM-GPFR | 6.47% | 6.43% | 6.61% | 6.62% | 6.78% | 6.6% | 6.71% | 6.58% | 8.41% | 8.14% | 7.92% | 11.52% | 10.48% | 7.44% | 6.77% |
| BHM-GPFR | 4.58% | 4.61% | 4.84% | 4.88% | 5.07% | 4.98% | 5.15% | 5.32% | 7.3% | 7.04% | 6.76% | 10.67% | 10.23% | 7.58% | 7.24% |

Figure 5: Multi-step prediction results of mix-GPFR, mix-GPNM, DPM-GPFR, HM-GPFR and BHM-GPFR.

be seen from the table that the prediction accuracy of HM-GPFR and BHM-GPFR is higher than that of mix-GPFR, mix-GPNM, and DPM-GPFR at almost every step, which shows that coarse-grained temporal information is helpful to improve the prediction performance, and the use of Markov chain to model the transfer law of electricity load evolution patterns can make effective use of coarse-grained temporal information.

Figure 5 further shows the results of multi-step prediction of these methods on the electricity load dataset. Here is also the case of "cold start" ($r = 1$), and we predict the electricity loads in the next 10 days (960 time points in total). It can be seen from the figure that these methods can effectively utilize the periodic structure in the time-series, and the prediction results show periodicity, but the prediction results of HM-GPFR and BHM-GPFR are slightly different from other methods. Due to the problem of "cold start", the predictions of mix-GPFR, mix-GPNM, and DPM-GPFR for each day are the same, *i.e.*, $\hat{\boldsymbol{y}}_{N+1} = \hat{\boldsymbol{y}}_{N+2} = \cdots = \hat{\boldsymbol{y}}_{N+10}$, while HM-GPFR and BHM-GPFR will use coarse-grained temporal information when making predictions, and then adjust the predicted values of each day. Based on the predicted values of other methods, it can be seen from the figure that the predicted values of HM-GPFR and BHM-GPFR on the first day are higher, and with the increase in step size, the predicted values will tend to the weighted average value of the mean function of each GPFR component.

## 6 CONCLUSION

In this paper, we have proposed the concept of multi-scale time series. Multi-scale time series have two granularity temporal structures. We established the HM-GPFR model for multi-scale time series forecasting and designed an effective learning algorithm. In addition, we also gave a priori to the parameters in the model, and obtain a more robust BHM-GPFR model. Compared with conventional GPFR-related methods (mix-GPFR, mix-GPNM, DPM-GPFR), the proposed method can effectively use the temporal information of both fine level and coarse level, alleviate the "cold start" problem, and has good performance in short-term prediction and long-term prediction. HM-GPFR and BHM-GPFR not only achieve high prediction accuracy but also have good interpretability. Combined with the actual problem background and domain knowledge, we can explain the state transition law learned by the model.

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

## A  Learning Algorithms of the Proposed Methods

### A.1  HM-GPFR

Due to the existence of latent variables $\{z_t\}_{t=1}^T$ , we apply the EM algorithm to learn the HM-GPFR model. We write $\mathcal{T} = \{\mathcal{D}_t\}_{t=1}^T$ to denote observations, $\boldsymbol{\Theta} = \{P_{kl}\}_{k,l=1}^K \cup \{\pi_k, \boldsymbol{b}_k, \boldsymbol{\theta}_k\}_{k=1}^K$ to denote all parameters, and $\boldsymbol{\Omega} = \{z_t\}_{t=1}^T$ to denote all latent variables. First, the complete data log-likelihood is

$$\mathcal{L}(\boldsymbol{\Theta}; \mathcal{T}, \boldsymbol{\Omega}) = \sum_{k=1}^K \mathbb{I}(z_1 = k) \log \boldsymbol{\pi}_k + \sum_{t=1}^{T-1} \sum_{k=1}^K \sum_{l=1}^K \mathbb{I}(z_{t+1} = l, z_t = k) \log P_{kl}$$

$$+ \sum_{t=1}^T \sum_{k=1}^K \mathbb{I}(z_t = k) \log \mathbb{P}(\boldsymbol{y}_t; \boldsymbol{b}_k, \boldsymbol{\theta}_k). \tag{10}$$

In the E-step of the EM algorithm, we need to calculate the expectation of Equation (10) with respect to the posterior distribution of latent variables $\boldsymbol{\Omega}$ to obtain the Q-function. However, it is not necessary to explicitly calculate $\mathbb{P}(\boldsymbol{\Omega}|\mathcal{T}; \boldsymbol{\Theta})$, which is a categorical distribution with $K^N$ possible values, and it suffices to obtain $\mathbb{P}(z_{t+1} = l, z_t = k|\mathcal{T}; \boldsymbol{\Theta})$ and $\mathbb{P}(z_t = k|\mathcal{T}; \boldsymbol{\Theta})$. We first introduce some notations as follows:

$$\alpha_t(k) = \mathbb{P}(\boldsymbol{y}_1, \boldsymbol{y}_2, \ldots, \boldsymbol{y}_t, z_t = k; \boldsymbol{\Theta}),$$
$$\beta_t(k) = \mathbb{P}(\boldsymbol{y}_{t+1}, \boldsymbol{y}_{t+2}, \ldots, \boldsymbol{y}_T | z_t = k; \boldsymbol{\Theta}),$$
$$\gamma_t(k) = \mathbb{P}(z_t = k|\mathcal{T}; \boldsymbol{\Theta}), \tag{11}$$
$$\xi_t(k, l) = \mathbb{P}(z_t = k, z_{t+1} = l|\mathcal{T}; \boldsymbol{\Theta}).$$

The key-point is to calculate $\gamma_t(k)$ and $\xi_t(k, l)$. Note that

$$\gamma_t(k) = \mathbb{P}(z_t = k|\mathcal{T}; \boldsymbol{\Theta}) \propto \mathbb{P}(z_t = k, \mathcal{T}; \boldsymbol{\Theta}) = \alpha_t(k)\beta_t(k),$$
$$\xi_t(k, l) = \alpha_t(k)P_{kl}\mathcal{N}(\boldsymbol{y}_{t+1}; \boldsymbol{\Phi}\boldsymbol{b}_l, \mathbf{C}_l)\beta_{t+1}(l). \tag{12}$$

Therefore, the problem boils down to calculate $\alpha_t(k)$ and $\beta_t(k)$. We can derive them recursively based on the forward-backward algorithm . According to the definition of $\alpha_t(k)$, we have

$$\alpha_1(k) = \pi_k \mathcal{N}(\boldsymbol{y}_1; \boldsymbol{\Phi}\boldsymbol{b}_k, \mathbf{C}_k) \quad , \quad \alpha_t(k) = \left( \sum_{l=1}^K \alpha_{t-1}(l)P_{lk} \right) \mathcal{N}(\boldsymbol{y}_t; \boldsymbol{\Phi}\boldsymbol{b}_k, \mathbf{C}_k). \tag{13}$$

Similarly, according to the definition of $\beta_t(k)$, we have

$$\beta_T(k) = 1 \quad , \quad \beta_t(k) = \sum_{l=1}^K P_{kl}\mathcal{N}(\boldsymbol{y}_{t+1}; \boldsymbol{\Phi}\boldsymbol{b}_l, \mathbf{C}_l)\beta_{t+1}(l). \tag{14}$$

To summary, in the E-step we first use Equations (13) and (14) to calculate $\alpha_t(k), \beta_t(k)$ recursively based on current parameters, then calculate $\gamma_t(k), \xi_t(k, l)$ according to Equation (12). The Q-function is given by

$$\mathcal{Q}(\boldsymbol{\Theta}) = \sum_{k=1}^K \gamma_1(k) \log \boldsymbol{\pi}_k + \sum_{t=1}^{T-1} \sum_{k=1}^K \sum_{l=1}^K \xi_t(k, l) \log P_{kl} + \sum_{t=1}^T \sum_{k=1}^K \gamma_t(k) \log \mathcal{N}(\boldsymbol{y}_t; \boldsymbol{\Phi}\boldsymbol{b}_k, \mathbf{C}_k). \tag{15}$$

In the M-step, we need to maximize $\mathcal{Q}$ with respect to parameters. The parameters $\{\pi_k\}_{k=1}^K$ and $\{P_{kl}\}_{k,l=1}^K$ can be optimized in closed form,

$$\pi_k = \frac{\gamma_1(k)}{\sum_{l=1}^K \gamma_1(l)} \quad , \quad P_{kl} = \frac{\sum_{t=1}^{T-1} \xi_t(k, l)}{\sum_{t=1}^{T-1} \sum_{m=1}^K \xi_t(k, m)}. \tag{16}$$

The parameters $\{\boldsymbol{b}_k, \boldsymbol{\theta}_k\}_{k=1}^K$ cannot be solved in closed form, and we apply the gradient ascent algorithm to optimize $\mathcal{Q}(\boldsymbol{\Theta})$ with gradients

$$\frac{\partial \mathcal{Q}(\boldsymbol{\Theta})}{\partial \boldsymbol{\theta}_k} = \frac{1}{2} \sum_{t=1}^T \gamma_t(k) \mathrm{tr} \left( (\mathbf{C}_k^{-1}(\boldsymbol{y}_t - \boldsymbol{\Phi}\boldsymbol{b}_k)(\boldsymbol{y}_t - \boldsymbol{\Phi}\boldsymbol{b}_k)^{\mathrm{T}} \mathbf{C}_k^{-1} - \mathbf{C}_k^{-1}) \frac{\partial \mathbf{C}_k}{\partial \boldsymbol{\theta}_k} \right),$$

$$\frac{\partial \mathcal{Q}(\boldsymbol{\Theta})}{\partial \boldsymbol{b}_k} = \sum_{t=1}^T \gamma_t(k) \boldsymbol{\Phi}^{\mathrm{T}} \mathbf{C}_k^{-1}(\boldsymbol{y}_t - \boldsymbol{\Phi}\boldsymbol{b}_k). \tag{17}$$

The complete algorithm is summarized in Algorithm 1. When the partial observations $y_{T+1,1}, \ldots, y_{T+1,M}$ become complete as we collect more data, we can adjust the parameters incrementally without retraining the model. This is achieved by continuing EM iterations with current parameters until the iteration converges again.

---

**Algorithm 1:** The EM algorithm for learning HM-GPFR.

---

Initialize parameters $\mathbf{\Theta}$;
**while** *not converged* **do**
    // E-step
1    $\alpha_1(k) = \pi_k \mathcal{N}(\boldsymbol{y}_1; \mathbf{\Phi}\boldsymbol{b}_k, \mathbf{C}_k)$;
2    **for** $t = 2, \ldots, T$ **do**
3        **for** $k = 1, 2, \ldots, K$ **do**
4            $\alpha_t(k) = \left( \sum_{l=1}^K \alpha_{t-1}(l) P_{lk} \right) \mathcal{N}(\boldsymbol{y}_t; \mathbf{\Phi}\boldsymbol{b}_k, \mathbf{C}_k)$;
5        **end**
6    **end**
7    $\beta_T = 1$;
8    **for** $t = T-1, \ldots, 1$ **do**
9        **for** $k = 1, 2, \ldots, K$ **do**
10           $\beta_t(k) = \sum_{l=1}^K P_{kl} \mathcal{N}(\boldsymbol{y}_{t+1}; \mathbf{\Phi}\boldsymbol{b}_l, \mathbf{C}_l)\beta_{t+1}(l)$;
11        **end**
12    **end**
13    **for** $t = 1, \ldots, T$ **do**
14        **for** $k = 1, 2, \ldots, K$ **do**
15           $\gamma_t(k) \propto \alpha_t(k)\beta_t(k)$;
16           **for** $l = 1, 2, \ldots, K$ **do**
17             $\xi_t(k, l) \propto \alpha_t(k) P_{kl} \mathcal{N}(\boldsymbol{y}_{t+1}; \mathbf{\Phi}\boldsymbol{b}_l, \mathbf{C}_l)\beta_{t+1}(l)$;
18           **end**
19           Normalize $\{\xi_t(k, l)\}_{l=1}^K$;
20        **end**
21        Normalize $\{\gamma_t(k)\}_{k=1}^K$;
22    **end**
    // M-step
23    **for** $k = 1, 2, \ldots, K$ **do**
24        $\pi_k = \frac{\gamma_1(k)}{\sum_{l=1}^K \gamma_1(l)}$;
25        **for** $l = 1, 2, \ldots, K$ **do**
26           $P_{kl} = \frac{\sum_{t=1}^{T-1} \xi_t(k,l)}{\sum_{t=1}^{T-1}\sum_{m=1}^K \xi_t(k,m)}$;
27        **end**
28    Using gradient ascent algorithm to optimize $\mathcal{Q}(\mathbf{\Theta})$ with respect to $\boldsymbol{\theta}_k$ and $\boldsymbol{b}_k$ according to Equation (17);
29    Update $\mathbf{C}_k$ with new parameters $\boldsymbol{\theta}_k$;
30    **end**
**end**

---

## A.2 BHM-GPFR

We still use the EM algorithm to learn the parameters of the BHM-GPFR model. However, this case is more complicated since there are more latent variables. The complete data log-likelihood is

$$
\begin{aligned}
\mathcal{L}(\boldsymbol{\Theta}; \mathcal{T}, \boldsymbol{\Omega}) = &\sum_{k=1}^{K} \log \mathcal{N}(\boldsymbol{b}_k; \mathbf{m}_b, \boldsymbol{\Sigma}_b) + \sum_{k=1}^{K} \sum_{l=1}^{K} (a_0 - 1) \log P_{kl} \\
&+ \sum_{k=1}^{K} \mathbb{I}(z_1 = k) \log \boldsymbol{\pi}_k + \sum_{t=1}^{T-1} \sum_{k=1}^{K} \sum_{l=1}^{K} \mathbb{I}(z_{t+1} = l, z_t = k) \log P_{kl} \\
&+ \sum_{t=1}^{T} \sum_{k=1}^{K} \mathbb{I}(z_t = k) \log \mathcal{N}(\boldsymbol{y}_t; \boldsymbol{\Phi}\boldsymbol{b}_k, \mathbf{C}_k).
\end{aligned} \tag{18}
$$

Compared with Equation (10), the first two terms are extra due to the prior distributions. In the E-step of EM algorithm, we need to take expectation of Equation (18) with respect to the posterior distribution of latent variables. However, the posterior of $\boldsymbol{\Omega}$ is intractable since $\{\boldsymbol{b}_k\}_{k=1}^{K}$, $\{\boldsymbol{p}_k\}_{k=1}^{K}$ and $\{z_t\}_{t=1}^{T}$ are correlated. We use the variational inference method and try to find an optimal approximation of $\mathbb{P}(\boldsymbol{\Omega}|\mathcal{T}; \boldsymbol{\Theta})$ with simple form. We adopt the mean-field family approximation, which factorizes the joint distribution of $\boldsymbol{\Omega}$ to a product of several independent distributions,

$$
\mathbb{Q}(\boldsymbol{\Omega}) = \prod_{k=1}^{K} \mathbb{Q}(\boldsymbol{b}_k) \prod_{k=1}^{K} \mathbb{Q}(\boldsymbol{p}_k) \mathbb{Q}(\boldsymbol{z}). \tag{19}
$$

Similar to the HM-GPFR case, $\mathbb{Q}(\boldsymbol{z})$ is a categorical distribution with $K^T$ possible values, but we do not need to calculate $\mathbb{Q}(\boldsymbol{z})$ explicitly and only need to calculate $\gamma_t(k) = \mathbb{Q}(z_t = k)$ and $\xi_t(k, l) = \mathbb{Q}(z_{t+1} = l, z_t = k)$. According to the variational inference theory, we iterate $\mathbb{Q}(\boldsymbol{b}_k)$, $\mathbb{Q}(\boldsymbol{p}_k)$ and $\mathbb{Q}(\boldsymbol{z})$ alternately until convergence.

For $\mathbb{Q}(\boldsymbol{b}_k)$,

$$
\begin{aligned}
\mathbb{Q}(\boldsymbol{b}_k) &\propto \exp \mathbb{E}_{\prod_{k=1}^{K} \mathbb{Q}(\boldsymbol{p}_k)\mathbb{Q}(\boldsymbol{z})}[\mathcal{L}(\boldsymbol{\Theta}; \mathcal{T}, \boldsymbol{\Omega})] \\
&= \exp \mathbb{E}_{\mathbb{Q}(\boldsymbol{z})} \left[ \log \mathcal{N}(\boldsymbol{b}_k; \mathbf{m}_b, \boldsymbol{\Sigma}_b) + \sum_{t=1}^{T} \mathbb{I}(z_t = k) \log \mathcal{N}(\boldsymbol{y}_t; \boldsymbol{\Phi}\boldsymbol{b}_k, \mathbf{C}_k) \right] \\
&\propto \exp \left( -\frac{1}{2} \log |\boldsymbol{\Sigma}_b| - \frac{1}{2} (\boldsymbol{b}_k - \mathbf{m}_b)^{\mathrm{T}} \boldsymbol{\Sigma}_b^{-1} (\boldsymbol{b}_k - \mathbf{m}_b) \right. \\
&\qquad \left. + \sum_{t=1}^{T} \gamma_t(k) \left( -\frac{1}{2} \log |\mathbf{C}_k| - \frac{1}{2} (\boldsymbol{y}_t - \boldsymbol{\Phi}\boldsymbol{b}_k)^{\mathrm{T}} \mathbf{C}_k^{-1} (\boldsymbol{y}_t - \boldsymbol{\Phi}\boldsymbol{b}_k) \right) \right)
\end{aligned} \tag{20}
$$

By completing the square, we obtain the approximate posterior of $\boldsymbol{b}_k$ is $\mathcal{N}(\mathbf{m}_k, \boldsymbol{\Sigma}_k)$ with

$$
\boldsymbol{\Sigma}_k = \left( \boldsymbol{\Sigma}_b + \sum_{t=1}^{T} \gamma_t(k) \boldsymbol{\Phi}^{\mathrm{T}} \mathbf{C}_k^{-1} \boldsymbol{\Phi} \right)^{-1}, \quad \mathbf{m}_k = \boldsymbol{\Sigma}_k \left( \boldsymbol{\Sigma}_b^{-1} \mathbf{m}_b + \sum_{t=1}^{T} \gamma_t(k) \boldsymbol{\Phi}^{\mathrm{T}} \mathbf{C}_k^{-1} \boldsymbol{y}_t \right). \tag{21}
$$

For $\mathbb{Q}(\boldsymbol{p}_k)$,

$$
\begin{aligned}
\mathbb{Q}(\boldsymbol{p}_k) &\propto \exp \mathbb{E}_{\prod_{k=1}^{K} \mathbb{Q}(\boldsymbol{b}_k)\mathbb{Q}(\boldsymbol{z})}[\mathcal{L}(\boldsymbol{\Theta}; \mathcal{T}, \boldsymbol{\Omega})] \\
&= \exp \mathbb{E}_{\mathbb{Q}(\boldsymbol{z})} \left[ \sum_{l=1}^{K} (a_0 - 1) \log P_{kl} + \sum_{t=1}^{T-1} \sum_{l=1}^{K} \mathbb{I}(z_{t+1} = l, z_t = k) \log P_{kl} \right] \\
&= \exp \left( \sum_{l=1}^{K} (a_0 - 1) \log P_{kl} + \sum_{t=1}^{T-1} \sum_{l=1}^{K} \xi_t(k, l) \log P_{kl} \right) \\
&= \prod_{l=1}^{K} P_{kl}^{a_0 + \sum_{t=1}^{T-1} \xi_t(k, l) - 1}
\end{aligned} \tag{22}
$$

Therefore, the approximate posterior of $\boldsymbol{p}_k$ is $\mathrm{Dir}(a_{k1}, \ldots, a_{kK})$ with $a_{kl} = a_0 + \sum_{t=1}^{T-1} \xi_t(k, l)$.

For $\mathbb{Q}(\boldsymbol{z})$,

$$
\begin{aligned}
\mathbb{Q}(\boldsymbol{z}) &\propto \exp \mathbb{E}_{\prod_{k=1}^K \mathbb{Q}(\boldsymbol{b}_k) \prod_{k=1}^K \mathbb{Q}(\boldsymbol{p}_k)}[\mathcal{L}(\boldsymbol{\Theta}; \mathcal{T}, \boldsymbol{\Omega})] \\
&= \exp \mathbb{E}_{\prod_{k=1}^K \mathbb{Q}(\boldsymbol{b}_k) \prod_{k=1}^K \mathbb{Q}(\boldsymbol{p}_k)} \left[ \sum_{k=1}^K \mathbb{I}(z_1 = k) \log \boldsymbol{\pi}_k + \sum_{t=1}^{T-1} \sum_{k=1}^K \sum_{l=1}^K \mathbb{I}(z_{t+1} = l, z_t = k) \log P_{kl} \right. \\
&\quad \left. + \sum_{t=1}^T \sum_{k=1}^K \mathbb{I}(z_t = k) \log \mathcal{N}(\boldsymbol{y}_t; \boldsymbol{\Phi}\boldsymbol{b}_k, \mathbf{C}_k) \right] \\
&= \exp \left( \sum_{k=1}^K \mathbb{I}(z_1 = k) \log \boldsymbol{\pi}_k + \sum_{t=1}^{T-1} \sum_{k=1}^K \sum_{l=1}^K \mathbb{I}(z_{t+1} = l, z_t = k) \mathbb{E}_{\mathbb{Q}(\boldsymbol{p}_k)}[\log P_{kl}] \right. \\
&\quad \left. + \sum_{t=1}^T \sum_{k=1}^K \mathbb{I}(z_t = k) \mathbb{E}_{\mathbb{Q}(\boldsymbol{b}_k)}[\log \mathcal{N}(\boldsymbol{y}_t; \boldsymbol{\Phi}\boldsymbol{b}_k, \mathbf{C}_k)] \right).
\end{aligned}
\tag{23}
$$

Note that this equation has exactly the same form as Equation (10), thus we can use the forward-backward algorithm to obtain $\gamma_t(k)$ and $\xi_t(k, l)$. To see this, let

$$
\tilde{P}_{kl} = \exp \mathbb{E}_{\mathbb{Q}(\boldsymbol{p}_k)}[\log P_{kl}] = \exp \left( \psi(a_{kl}) - \psi(\sum_{l=1}^K a_{kl}) \right),
$$

$$
\tilde{\mathbb{P}}(\boldsymbol{y}_t; \mathbf{m}_k, \boldsymbol{\Sigma}_k, \boldsymbol{\theta}_k) = \exp \mathbb{E}_{\mathbb{Q}(\boldsymbol{b}_k)}[\log \mathcal{N}(\boldsymbol{y}_t; \boldsymbol{\Phi}\boldsymbol{b}_k, \mathbf{C}_k)] = \mathcal{N}(\boldsymbol{y}_t; \boldsymbol{\Phi}\mathbf{m}_k, \mathbf{C}_k) \exp \left( -\frac{1}{2} \mathrm{tr}(\boldsymbol{\Sigma}_k \boldsymbol{\Phi}\mathbf{C}_k^{-1}\boldsymbol{\Phi}^{\mathrm{T}}) \right),
\tag{24}
$$

then Equation (23) can be rewritten as

$$
\begin{aligned}
\log \mathbb{Q}(\boldsymbol{z}) &= \sum_{k=1}^K \mathbb{I}(z_1 = k) \log \boldsymbol{\pi}_k + \sum_{t=1}^{T-1} \sum_{k=1}^K \sum_{l=1}^K \mathbb{I}(z_{t+1} = l, z_t = k) \log \tilde{P}_{kl} \\
&\quad + \sum_{t=1}^T \sum_{k=1}^K \mathbb{I}(z_t = k) \log \tilde{\mathbb{P}}(\boldsymbol{y}_t; \mathbf{m}_k, \boldsymbol{\Sigma}_k, \boldsymbol{\theta}_k).
\end{aligned}
\tag{25}
$$

To obtain $\gamma_t(k)$ and $\xi_t(k, l)$, we run the Baum-Welch algorithm with sufficient statistics $\boldsymbol{\pi}_k, \tilde{P}_{kl}, \tilde{\mathbb{P}}(\boldsymbol{y}_t; \mathbf{m}_k, \boldsymbol{\Sigma}_k, \boldsymbol{\theta})$.

Taking expectation of Equation (18) with respect to the approximate posterior $\mathbb{Q}(\boldsymbol{\Omega})$, the Q function is

$$
\begin{aligned}
\mathcal{Q}(\boldsymbol{\Theta}) &= \sum_{k=1}^K \mathbb{E}_{\mathbb{Q}(\boldsymbol{b}_k)}[\log \mathcal{N}(\boldsymbol{b}_k; \mathbf{m}_b, \boldsymbol{\Sigma}_b)] + \sum_{k=1}^K \gamma_1(k) \log \boldsymbol{\pi}_k + \sum_{t=1}^T \sum_{k=1}^K \gamma_t(k) \mathbb{E}_{\mathbb{Q}(\boldsymbol{b}_k)}[\log \mathcal{N}(\boldsymbol{y}_t; \boldsymbol{\Phi}\boldsymbol{b}_k, \mathbf{C}_k)] \\
&= \sum_{k=1}^K (\log \mathcal{N}(\mathbf{m}_k; \mathbf{m}_b, \boldsymbol{\Sigma}_b) - \frac{1}{2} \mathrm{tr}(\boldsymbol{\Sigma}_k \boldsymbol{\Sigma}_b^{-1})) + \sum_{k=1}^K \gamma_1(k) \log \boldsymbol{\pi}_k \\
&\quad + \sum_{t=1}^T \sum_{k=1}^K \gamma_t(k) (\log \mathcal{N}(\boldsymbol{y}_t; \boldsymbol{\Phi}\mathbf{m}_k, \mathbf{C}_k) - \frac{1}{2} \mathrm{tr}(\boldsymbol{\Sigma}_k \boldsymbol{\Phi}\mathbf{C}_i^{-1}\boldsymbol{\Phi}^{\mathrm{T}}))].
\end{aligned}
\tag{26}
$$

Maximizing $\mathcal{Q}(\boldsymbol{\Theta})$ with respect to $\pi_k$, $\mathbf{m}_b$ and $\boldsymbol{\Sigma}_b$, we obtain

$$
\pi_k = \frac{\gamma_1(k)}{\sum_{l=1}^K \gamma_1(l)} \quad , \quad \boldsymbol{\Sigma}_b = \frac{1}{K} \sum_{k=1}^K \left( \boldsymbol{\Sigma}_k + (\mathbf{m}_k - \mathbf{m}_b)(\mathbf{m}_k - \mathbf{m}_b)^{\mathrm{T}} \right) \quad , \quad \mathbf{m}_b = \frac{1}{K} \sum_{k=1}^K \mathbf{m}_k.
\tag{27}
$$

The parameters $\{\boldsymbol{\theta}_k\}_{k=1}^K$ cannot be solved in closed form, and we apply the gradient ascent algorithm to optimize $\mathcal{Q}(\boldsymbol{\Theta})$. The gradient of $\mathcal{Q}(\boldsymbol{\Theta})$ with respect to $\boldsymbol{\theta}_k$ is

$$\frac{\partial \mathcal{Q}(\boldsymbol{\Theta})}{\partial \boldsymbol{\theta}_k} = \sum_{t=1}^T \frac{1}{2} \gamma_t(k) \mathrm{tr}\left(\mathbf{C}_k^{-1}\mathbf{S}_{t,k}\mathbf{C}_k^{-1}\frac{\partial \mathbf{C}_k}{\partial \boldsymbol{\theta}_k}\right), \mathbf{S}_{t,k} = (\boldsymbol{y}_t-\boldsymbol{\Phi}\mathbf{m}_k)(\boldsymbol{y}_t-\boldsymbol{\Phi}\mathbf{m}_k)^{\mathrm{T}}+\boldsymbol{\Phi}^{\mathrm{T}}\boldsymbol{\Sigma}_k\boldsymbol{\Phi}-\mathbf{C}_k \,. \tag{28}$$

The complete algorithm is summarized in Algorithm 2.

## B  MORED EXPERIMENTAL RESULTS

### B.1  DETAILED EXPERIMENT SETTINGS

For AR, MA, ARMA, ARIMA, and SARMA, we set the model order $L$ in $\{4, 8, 12\}$. For SARMA, the seasonal length is set to be 96 since there are 96 records per day, which implicitly assumes that the overall time-series exhibits periodicity in days. LSTM, NN, SVR, and EGPM transform the time-series prediction problem into a regression problem, $i.e.$, use the latest $L$ observations to predict the output at the next point and then use the regression method to train and predict. In the experiment, we set $L$ in $\{4, 12, 24, 48\}$. The neural network in the FNN has two hidden layers with 10 and 5 neurons, respectively. The kernel function in SVR is the Gaussian kernel whose scale parameters are adaptively selected by cross-validation. The number of components for EGPM is set in $\{3, 5, 10\}$. In addition, we use the recursive method **?** for multi-step prediction. For mix-GPFR, mix-GPNM, and DPM-GPFR, we first convert the time-series data into curve datasets and then use these methods to make predictions. The number of components $K$ in mix-GPFR and mix-GPNM is set to 5 and the number of B-spline basis functions $D$ in mix-GPFR and DPM-GPFR is set to 30.

### B.2  CLUSTERING STRUCTURE

Estimated values of latent variable $\hat{z}_i$ also indicate the evolution mode corresponding to the data of the $i$-th day. Figure 6 visualizes some training data with different colors indicating different evolution modes, so we can intuitively see the multi-scale structure in the electricity load time-series. According to the learned transition probability, we can obtain the stationary distribution of Markov chain $(z_1, z_2, \ldots, z_N)$, which is $[0.4825, 0.2026, 0.0513, 0.1124, 0.1513]^{\mathrm{T}}$ in HM-GPFR, and $[0.4501, 0.0427, 0.2992, 0.1381, 0.0700]^{\mathrm{T}}$ in BHM-GPFR. The proportion of each mode in fig. 6 is roughly consistent with the stationary distribution.

### B.3  MULTI-STEP PREDICTION UNDER COLD-START SETTING

In order to more clearly see the role of Markov chain structure of hidden variables in the cold start setting, in Figure 7 and Figure 8, we show the predicted values of HM-GPFR and BHM-GFPR for electricity load in the next five days $\hat{\boldsymbol{y}}_{N+1}, \ldots, \hat{\boldsymbol{y}}_{N+5}$ and distributions of latent variables $z_{N+1}, \ldots, z_{N+5}$ conditioned on $\hat{z}_N = k$. It can be seen from the figure that HM-GPFR and BHM-GPFR have different predictions for each day's electricity load, which will be adjusted according to the transition probability of evolution law. For example, in Figure 7, when $\hat{z}_N = 1$, the power load on that day is low, and the predicted value of HM-GPFR on the $(N+1)$-th day is also low. When $hatz_N = 5$, the electricity load on that day is higher, and the predicted value of HM-GPFR on the $(N+1)$-th day is also higher. Figure 8 has a similar phenomenon. In addition, it can be seen that with the increase of $i_*$, $\mathbb{P}(z_{i_*})$ quickly converges to the stable distribution of the Markov chain, and the predicted value $\hat{\boldsymbol{y}}_{i_*}$ also tends to be the weighted average of the mean function in each GPFR component. In conclusion, these phenomena demonstrate that HM-GPFR and BHM-GPFR can effectively use the coarse-grained temporal structure to adjust the prediction of each day.

### B.4  SENSITIVITY OF HYPER-PARAMETERS

There are two main hyper-parameters in HM-GPFR and BHM-GPFR: the number of B-spline basis functions $D$ and the number of GPFR components $K$. Here we mainly focus on the selection of $K$. We vary $K$ in $\{3, 4, 5, 6, 7, 8, 9, 10, 15, 20, 30, 50\}$, train HM-GPFR and BHM-GPFR respectively, and report the results in table 3. For HM-GPFR, its prediction performance tends to deteriorate with

---

**Algorithm 2:** The Variational EM algorithm for learning BHM-GPFR.

---

Initialize parameters $\boldsymbol{\Theta}$;

**while** *not converged* **do**

    // Variational E-step

1    Initialize variational parameters $\{\mathbf{m}_k, \boldsymbol{\Sigma}_k, a_{k1}, a_{k2}, \ldots, a_{kK}\}_{k=1}^{K}$. **while** *not converged* **do**

        // Calculate surrogate parameters.

2        **for** $k = 1, 2, \ldots, K$ **do**

3            $\tilde{\mathbb{P}}(\boldsymbol{y}_t; \mathbf{m}_k, \boldsymbol{\Sigma}_k, \boldsymbol{\theta}_k) = \mathcal{N}(\boldsymbol{y}_t; \boldsymbol{\Phi}\mathbf{m}_k, \mathbf{C}_k) \exp\left(-\frac{1}{2}\text{tr}(\boldsymbol{\Sigma}_k \boldsymbol{\Phi} \mathbf{C}_k^{-1} \boldsymbol{\Phi}^{\mathrm{T}})\right)$;

4            **for** $l = 1, 2, \ldots, K$ **do**

5                $\tilde{P}_{kl} = \exp\left(\psi(a_{kl}) - \psi(\sum_{l=1}^{K} a_{kl})\right)$;

6            **end**

7        **end**

        // Forward-backward algorithm

8        $\alpha_1(k) = \pi_k \tilde{\mathbb{P}}(\boldsymbol{y}_1; \boldsymbol{b}_k, \boldsymbol{\theta}_k)$;

9        **for** $t = 2, \ldots, T$ **do**

10       **for** $k = 1, 2, \ldots, K$ **do**

11          $\alpha_t(k) = \left(\sum_{l=1}^{K} \alpha_{t-1}(l) \tilde{P}_{lk}\right) \tilde{\mathbb{P}}(\boldsymbol{y}_t; \mathbf{m}_k, \boldsymbol{\Sigma}_k, \boldsymbol{\theta}_k)$;

12       **end**

13       **end**

14       $\beta_T = 1$;

15       **for** $t = T - 1, \ldots, 1$ **do**

16       **for** $k = 1, 2, \ldots, K$ **do**

17          $\beta_t(k) = \sum_{l=1}^{K} \tilde{P}_{kl} \tilde{\mathbb{P}}(\boldsymbol{y}_{t+1}; \mathbf{m}_l, \boldsymbol{\Sigma}_l, \boldsymbol{\theta}_l) \beta_{t+1}(l)$;

18       **end**

19       **end**

20       **for** $t = 1, \ldots, T$ **do**

21       **for** $k = 1, 2, \ldots, K$ **do**

22          $\gamma_t(k) \propto \alpha_t(k) \beta_t(k)$;

23          **for** $l = 1, 2, \ldots, K$ **do**

24              $\xi_t(k, l) \propto \alpha_t(k) \tilde{P}_{kl} \tilde{\mathbb{P}}(\boldsymbol{y}_{t+1}; \mathbf{m}_l, \boldsymbol{\Sigma}_l, \boldsymbol{\theta}_l) \beta_{t+1}(l)$;

25          **end**

26          Normalize $\{\xi_t(k, l)\}_{l=1}^{K}$;

27       **end**

28       Normalize $\{\gamma_t(k)\}_{k=1}^{K}$;

29       **end**

        // Update posterior $\mathbb{Q}(\boldsymbol{b}_k)$ and $\mathbb{Q}(\boldsymbol{p}_k)$.

30       **for** $k = 1, 2, \ldots, K$ **do**

31          $\boldsymbol{\Sigma}_k = \left(\boldsymbol{\Sigma}_b + \sum_{t=1}^{T} \gamma_t(k) \boldsymbol{\Phi}^{\mathrm{T}} \mathbf{C}_k^{-1} \boldsymbol{\Phi}\right)^{-1}$;

32          $\mathbf{m}_k = \boldsymbol{\Sigma}_k \left(\boldsymbol{\Sigma}_b^{-1} \mathbf{m}_b + \sum_{t=1}^{T} \gamma_t(k) \boldsymbol{\Phi}^{\mathrm{T}} \mathbf{C}_k^{-1} \boldsymbol{y}_t\right)$;

33          **for** $l = 1, 2, \ldots, K$ **do**

34              $a_{kl} = a_0 + \sum_{t=1}^{T-1} \xi_t(k, l)$;

35          **end**

36       **end**

37    **end**

    // M-step

38    $\boldsymbol{\Sigma}_b = \frac{1}{K} \sum_{k=1}^{K} \left(\boldsymbol{\Sigma}_k + (\mathbf{m}_k - \mathbf{m}_b)(\mathbf{m}_k - \mathbf{m}_b)^{\mathrm{T}}\right)$ , $\mathbf{m}_b = \frac{1}{K} \sum_{k=1}^{K} \mathbf{m}_k$;

39    **for** $k = 1, 2, \ldots, K$ **do**

40        $\pi_k = \frac{\gamma_1(k)}{\sum_{l=1}^{K} \gamma_1(l)}$;

41        Using gradient ascent algorithm to optimize $\mathcal{Q}(\boldsymbol{\Theta})$ with respect to $\boldsymbol{\theta}_k$ according to Equation (28);

42        Update $\mathbf{C}_k$ with new parameters $\boldsymbol{\theta}_k$;

43    **end**

**end**

---

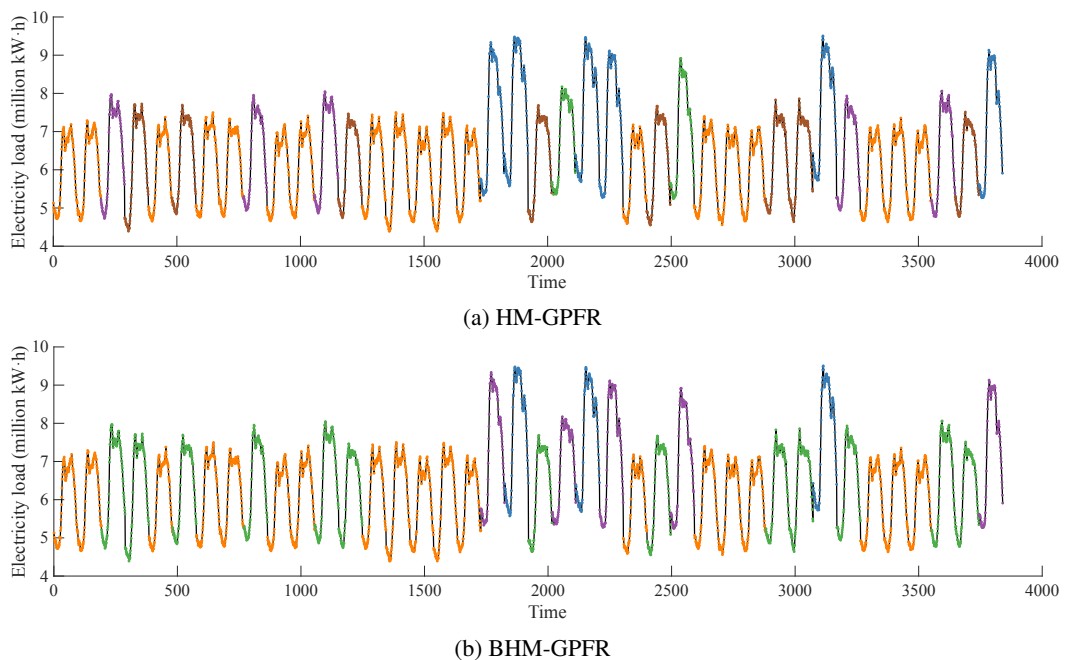

(a) HM-GPFR

(b) BHM-GPFR

Figure 6: Training time-series are divided into different evolving laws based on the learning results of HM-GPFR and BHM-GPFR.

Table 3: Sensitivity of HM-GPFR and BHM-GPFR with respect to the number of components $K$.

| Method | $K$ | Step length $S$ | | | | | | | | | | | | | | |
|---|---|---|---|---|---|---|---|---|---|---|---|---|---|---|---|---|
| | | 1 | 2 | 3 | 4 | 5 | 10 | 20 | 30 | 50 | 80 | 100 | 200 | 300 | 500 | 1000 |
| HM-GPFR | 3 | 0.84% | 1.01% | 1.19% | 1.35% | 1.52% | 2.35% | 3.85% | 4.82% | 6.36% | 8.48% | 9.63% | 10.71% | 9.43% | 6.78% | 6.75% |
| | 4 | 0.9% | 1.09% | 1.27% | 1.44% | 1.61% | 2.46% | 4.02% | 5.09% | 6.6% | 8.57% | 9.68% | 10.71% | 9.43% | 6.78% | 6.75% |
| | 5 | 0.93% | 1.12% | 1.3% | 1.48% | 1.66% | 2.51% | 4.07% | 5.18% | 6.79% | 8.8% | 9.83% | 10.76% | 9.49% | 6.82% | 6.77% |
| | 6 | 1.09% | 1.32% | 1.57% | 1.81% | 2.04% | 3.13% | 4.77% | 5.83% | 7.3% | 9.04% | 10.03% | 10.9% | 9.59% | 6.88% | 6.8% |
| | 7 | 1.06% | 1.27% | 1.49% | 1.7% | 1.91% | 2.87% | 4.5% | 5.66% | 7.28% | 9.17% | 10.13% | 10.88% | 9.58% | 6.87% | 6.79% |
| | 8 | 0.97% | 1.16% | 1.36% | 1.54% | 1.73% | 2.57% | 4.14% | 5.32% | 6.99% | 8.95% | 9.92% | 10.81% | 9.57% | 6.88% | 6.79% |
| | 9 | 1.1% | 1.33% | 1.56% | 1.79% | 2.01% | 3.06% | 4.77% | 5.93% | 7.49% | 9.3% | 10.29% | 10.99% | 9.6% | 6.88% | 6.8% |
| | 10 | 1.18% | 1.41% | 1.65% | 1.88% | 2.11% | 3.22% | 4.97% | 6.05% | 7.53% | 9.39% | 10.45% | 11.17% | 9.71% | 6.95% | 6.83% |
| | 15 | 1.25% | 1.48% | 1.72% | 1.94% | 2.17% | 3.29% | 5.03% | 6.12% | 7.63% | 9.42% | 10.5% | 11.22% | 9.71% | 6.94% | 6.82% |
| | 20 | 1.31% | 1.54% | 1.77% | 2.0% | 2.22% | 3.33% | 5.07% | 6.14% | 7.65% | 9.57% | 10.78% | 11.52% | 9.85% | 7.01% | 6.86% |
| | 30 | 1.37% | 1.62% | 1.87% | 2.12% | 2.38% | 3.62% | 5.45% | 6.5% | 7.98% | 9.76% | 10.92% | 11.57% | 9.86% | 7.01% | 6.86% |
| | 50 | 1.47% | 1.72% | 1.99% | 2.25% | 2.5% | 3.7% | 5.5% | 6.62% | 8.29% | 10.35% | 11.66% | 12.01% | 10.06% | 7.12% | 6.91% |
| BHM-GPFR | 3 | 0.85% | 1.02% | 1.19% | 1.36% | 1.52% | 2.35% | 3.87% | 4.85% | 6.37% | 8.46% | 9.6% | 10.7% | 9.47% | 6.84% | 6.82% |
| | 4 | 0.78% | 0.93% | 1.07% | 1.18% | 1.29% | 1.86% | 2.82% | 3.49% | 4.8% | 6.96% | 8.23% | 9.91% | 9.04% | 6.68% | 6.85% |
| | 5 | 0.77% | 0.92% | 1.07% | 1.18% | 1.3% | 1.89% | 2.88% | 3.59% | 4.89% | 6.88% | 8.04% | 9.85% | 9.21% | 6.94% | 7.15% |
| | 6 | 0.8% | 0.96% | 1.1% | 1.23% | 1.36% | 2.02% | 3.17% | 3.97% | 5.32% | 7.22% | 8.32% | 9.91% | 9.3% | 7.01% | 7.18% |
| | 7 | 0.79% | 0.95% | 1.1% | 1.22% | 1.33% | 1.94% | 3.01% | 3.79% | 5.12% | 6.89% | 7.99% | 9.76% | 9.34% | 7.18% | 7.39% |
| | 8 | 0.78% | 0.94% | 1.08% | 1.19% | 1.31% | 1.89% | 2.94% | 3.71% | 5.03% | 6.74% | 7.79% | 9.7% | 9.49% | 7.46% | 7.7% |
| | 9 | 0.78% | 0.93% | 1.07% | 1.18% | 1.29% | 1.86% | 2.86% | 3.61% | 4.92% | 6.69% | 7.77% | 9.73% | 9.52% | 7.53% | 7.8% |
| | 10 | 0.82% | 0.98% | 1.13% | 1.26% | 1.4% | 2.11% | 3.29% | 4.09% | 5.37% | 7.01% | 8.04% | 9.94% | 9.8% | 7.86% | 8.12% |
| | 15 | 0.79% | 0.94% | 1.07% | 1.18% | 1.29% | 1.84% | 2.86% | 3.64% | 4.95% | 6.66% | 7.7% | 9.89% | 9.96% | 8.25% | 8.6% |
| | 20 | 0.79% | 0.94% | 1.07% | 1.17% | 1.28% | 1.83% | 2.83% | 3.6% | 4.88% | 6.51% | 7.5% | 9.95% | 10.32% | 8.88% | 9.31% |
| | 30 | 0.8% | 0.95% | 1.07% | 1.18% | 1.29% | 1.83% | 2.82% | 3.58% | 4.86% | 6.52% | 7.53% | 10.04% | 10.46% | 9.07% | 9.52% |
| | 50 | 0.83% | 0.98% | 1.11% | 1.22% | 1.33% | 1.88% | 2.9% | 3.68% | 4.96% | 6.5% | 7.46% | 10.14% | 10.71% | 9.45% | 9.9% |

the increase of $K$. In short-term prediction, MAPE increases significantly, while MAPE changes less in long-term prediction. With the increase of $K$, the number of parameters in the model also increases, and the model tends to suffer from over-fitting. For BHM-GPFR, with the increase of $K$, its long-term prediction performance decreases significantly, while the medium-term and short-term prediction results do not change much. This shows that BHM-GPFR can prevent overfitting to a certain extent after introducing prior distributions to parameters. In addition, we also note that when $K \leq 10$, the difference between the results corresponding to different $K$ is not significant, which is a more reasonable choice. From the perspective of application, we set $K = 5$ in the experiment, which can take both expression ability and interpretability of the model into consideration.

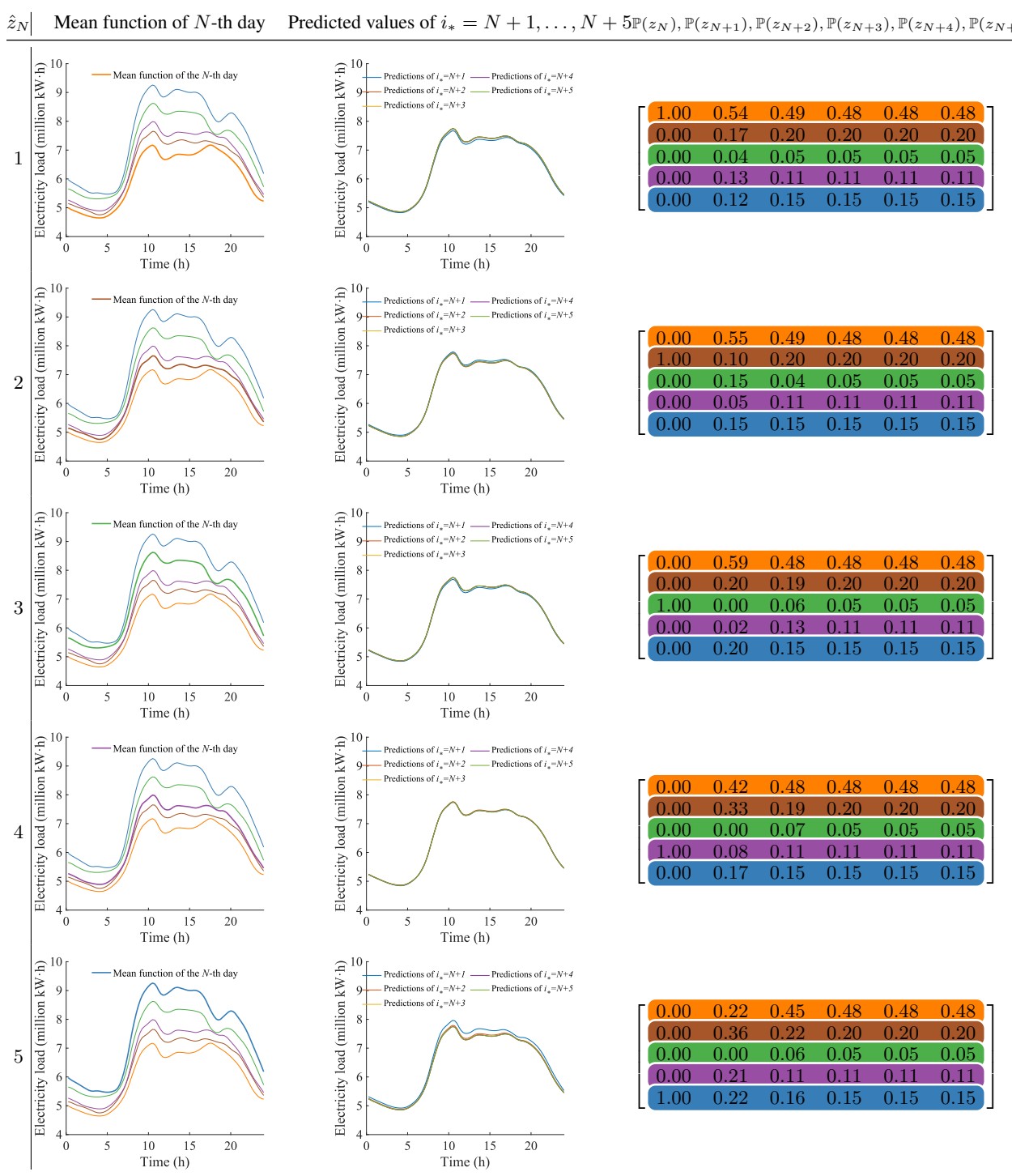

Figure 7: Estimated values $\hat{\boldsymbol{y}}_{N+1}, \ldots, \hat{\boldsymbol{y}}_{N+5}$ and distributions of $z_N, \ldots, z_{N+5}$ of HM-GPFR under $\hat{z} = k$ where $k = 1, 2, 3, 4, 5$.

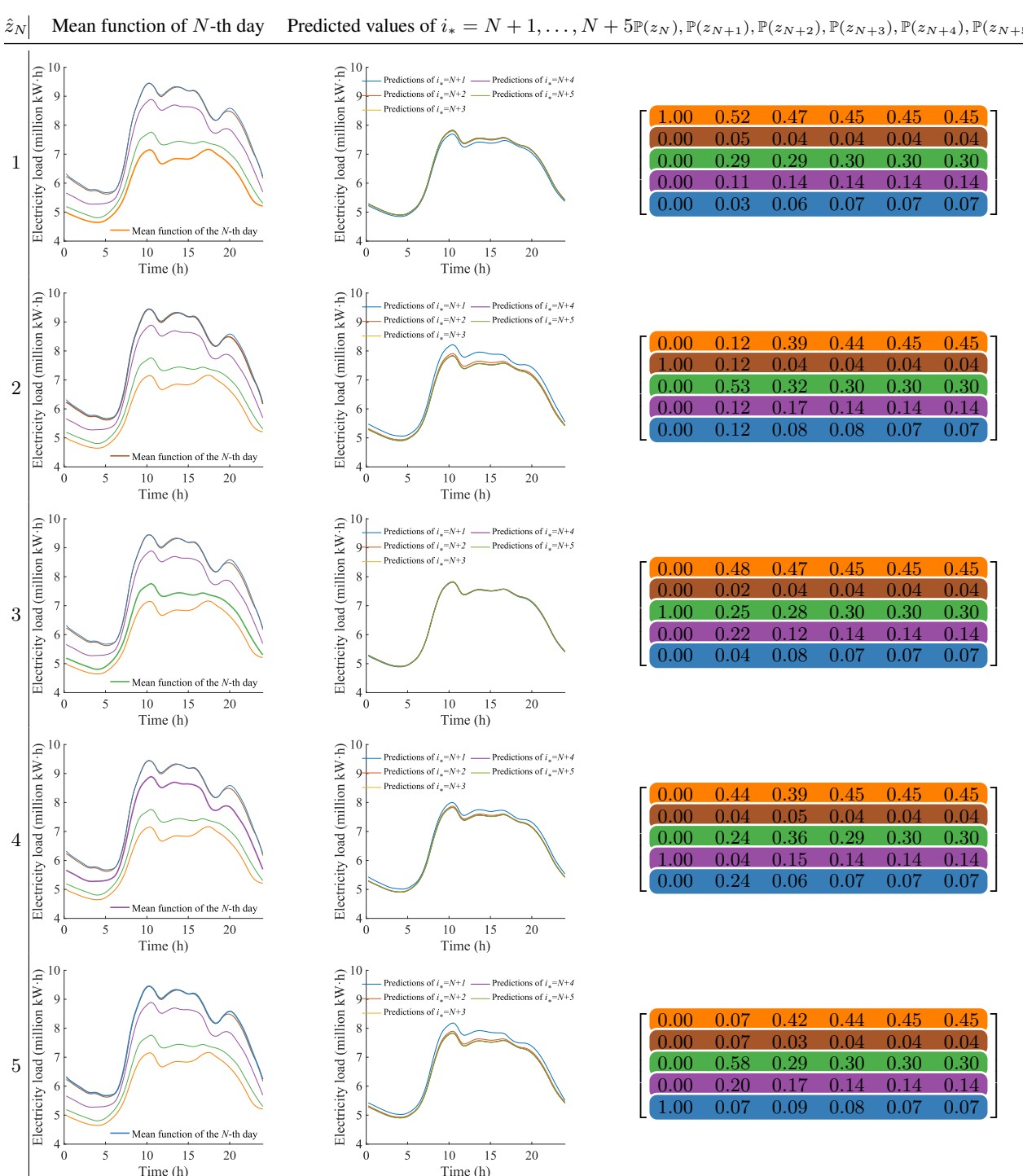

Figure 8: Estimated values $\hat{\boldsymbol{y}}_{N+1}, \ldots, \hat{\boldsymbol{y}}_{N+5}$ and distributions of $z_N, \ldots, z_{N+5}$ of BHM-GPFR under $\hat{z} = k$ where $k = 1, 2, 3, 4, 5$.

