# OpenReview forum: "Hidden Markov Mixture of Gaussian Process Functional Regression: Utilizing Multi-Scale Structure for Time-Series Forecasting"
_ICLR.cc/2023/Conference — Submitted to ICLR 2023_

### Official Review · Reviewer_CSZt · 2022-10-23

**Confidence:** 4
**Correctness:** 3
**Technical Novelty And Significance:** 2
**Empirical Novelty And Significance:** 2
**Recommendation:** 3

**Clarity, Quality, Novelty And Reproducibility:**

Clarity: The paper is easy to follow, but not self-contained.

Quality: The derivation of inference seems correct.

Novelty: The submission is lacking in novelty.

Reproducibility: I did not check the code.

**Strength And Weaknesses:**

Strength: The paper is easy to follow and the derivation of inference seems correct but I did not check it carefully.

Weakness: The submission is lacking in novelty and not self-contained.

**Summary Of The Paper:**

The submission proposed the hidden Markov based GPFR mixture model (HM-GPFR) by describing the time series data from the perspective of both fine and coarse level. The time series data at fine level is characterized by Gaussian process model and and that at coarse level is characterized by a hidden Markov process. To avoid overfitting, the work place priori on the model parameters and develop a Bayesian version.

**Summary Of The Review:**

The major concern of the submission is novelty. All key ideas of the proposed method: gaussian process regression, hidden Markov model, and EM (forward-backward) algorithm, are all existing in previous works. There is nothing new in the methodology in this submission. Another minor conceren is that the submission is not self-contained very well; it is confusing to see some notations without definition, e.g., $\gamma_t(k)$ above Eq.(7) and below Eq. (9), $m_k$ and $a_{kl}$ below Eq.(9). I guess these may be defined in the appendix, but the main text should be written in a self-contained way.

---

### Official Review · Reviewer_tgKE · 2022-10-24

**Confidence:** 3
**Correctness:** 3
**Technical Novelty And Significance:** 2
**Empirical Novelty And Significance:** 3
**Recommendation:** 5

**Clarity, Quality, Novelty And Reproducibility:**

The paper is clearly written overall, and the combination of multiple ideas from the literature is novel and likely relevant. Reproducibility, however, is poor since settings for the baselines are not properly discussed.

**Strength And Weaknesses:**

Strengths:
-- The model combines some existing ideas from the literature into one complex model which is able to deal with multi-scale time series.
-- A good level of interpretability is next to good performance.
Weaknesses:
-- The structure of the model is fairly complex, and modes of failure are not completely clear (i.e. what are the limitations of the proposed approach).
-- I would expect the inference method to be fairly slow and slower than competitors; there is, unfortunately, no comparison regarding computational costs.
-- Details regarding the specifications of the competitors are not clear. For example, did ARIMA include a seasonal component, how exactly was LSTM specified and trained, what was the choice of the kernels in GP-based approaches etc.?

Minor: it would be useful if Table 1 and Table 3 highlighted the best-performing methods to get an overall picture better.



**Summary Of The Paper:**

The paper proposes an interesting method for multi-scale time series based on Hidden Markov mixture models and Gaussian processes. Overall, the model is capable of fitting the data with switching dynamics, and experiments demonstrate good performance compared to the baselines.

**Summary Of The Review:**

The paper proposes an interesting model but, overall, lacks reproducibility details which are necessary for a publication. The settings of the baseline methods are not discussed in sufficient detail. Based on this, I cannot recommend acceptance at the current moment.

---

### Official Review · Reviewer_hJkz · 2022-10-25

**Confidence:** 4
**Correctness:** 3
**Technical Novelty And Significance:** 2
**Empirical Novelty And Significance:** 2
**Recommendation:** 5

**Clarity, Quality, Novelty And Reproducibility:**

The paper is well-written and easy to follow. The novelty of the paper is a bit limited and the numerical support of the proposed method is not enough convincing.

**Strength And Weaknesses:**

Strength: the paper is well-written and easy to follow.
Weakness: the proposed method is only a straightforward extension of the existing literature, e.g., [1]. The only difference is to introduce of the correlation between the indicator variables via a Markov chain. The benefits of this modification, however, are not well justified.

1. The performance of the proposed method is only evaluated through one data set and results are not very impressive compared to existing methods (mix-GPFR, DPM-GPFR). In addition, the real data has too many irregularities and it is hard to grasp the idea of what are the advantages and limitations of the proposed method. In my opinion, this can be better achieved through carefully designed synthetic experiments, especially considering that the work is purely empirical.

2. In section 5.1, it is probably better to consider SARIMA model compared to the SARMA model because the process is clearly nonstationary even after taking into account the seasonality.

3. In section 5.3, HM-GPFR and BHM-GPFR perform better when $r=1$. Can you study how the performances change as $r$ gradually increases? This perhaps will give more insights into why the proposed method work.

4. In equation (6), the transition probability only depends on the previous state, which may not be very suitable for grid data. It may be a better idea to include other coarse-level covariates such as predicted temperature, day of the week, etc.

5. The claim on page 2, "while HM-GPFR can adjust the parameters incrementally without retraining the model ", is not correct. The model parameters still need to be updated when new data are available.

Reference:

[1] Shi, Jian Qing, and B. Wang. "Curve prediction and clustering with mixtures of Gaussian process functional regression models." Statistics and Computing 18, no. 3 (2008): 267-283.

**Summary Of The Paper:**

This paper proposes a mixture Gaussian process functional regression model coupled with a hidden Markov chain at the coarse level. A Gaussian process regression is applied to the intra-day time series and then inter-day correlations are modeled through the hidden Markov chain. A Bayes version (BHM-GPFR) of the proposed model (HM-GPFR) is proposed to further improve the robustness of the method. The effectiveness of the proposed is demonstrated through an application to the electricity load data.

**Summary Of The Review:**

The paper is well-written and easy to follow. The novelty of the paper is a bit limited and the numerical support of the proposed method is not enough convincing.

---

### Decision · Program_Chairs · 2023-01-20

**Decision:**

Reject

**Justification For Why Not Higher Score:**

The paper is a straightforward modification of the existing literature and all the reviewers were aligned in rejecting the paper.

**Justification For Why Not Lower Score:**

N/A

**Metareview: Summary, Strengths And Weaknesses:**

The reviewers enjoyed how easy it was to follow your paper and provided good summaries of your work in their initial assessment. The main strength of your work is how well is written. At the same time, the model can be understood as a direct extension of the existing literature (see the first review). Also, the performance of the proposed model is not significantly different from the compared model. For a way forward, besides looking at the reference proposed by Reviewer hJkz, I suggest you look for other datasets in which your method would provide an edge before resubmitting.

**Summary Of Ac-Reviewer Meeting:**

This was not a borderline paper.